# On Disentangled Training for Nonlinear Transform in Learned Image Compression

**Han Li[1], Shaohui Li[2]\*, Wenrui Dai[1]\*, Maida Cao[1], Nuowen Kan[1], Chenglin Li[1], Junni Zou[1], Hongkai Xiong[1]**

[1]Shanghai Jiao Tong University, [2]Tsinghua Shenzhen International Graduate School, Tsinghua University

`{qingshi9974,daiwenrui}@sjtu.edu, lishaohui@sz.tsinghua.edu.cn`

## Abstract

Learned image compression (LIC) has demonstrated superior rate-distortion (R-D) performance compared to traditional codecs, but is challenged by training inefficiency that could incur more than two weeks to train a state-of-the-art model from scratch. Existing LIC methods overlook the slow convergence caused by compacting energy in learning nonlinear transforms. In this paper, we first reveal that such energy compaction consists of two components, *i.e.*, feature decorrelation and uneven energy modulation. On such basis, we propose a linear auxiliary transform (AuxT) to disentangle energy compaction in training nonlinear transforms. The proposed AuxT obtains coarse approximation to achieve efficient energy compaction such that distribution fitting with the nonlinear transforms can be simplified to fine details. We then develop wavelet-based linear shortcuts (WLSs) for AuxT that leverages wavelet-based downsampling and orthogonal linear projection for feature decorrelation and subband-aware scaling for uneven energy modulation. AuxT is lightweight and plug-and-play to be integrated into diverse LIC models to address the slow convergence issue. Experimental results demonstrate that the proposed approach can accelerate training of LIC models by 2 times and simultaneously achieves an average 1% BD-rate reduction. To our best knowledge, this is one of the first successful attempt that can significantly improve the convergence of LIC with comparable or superior rate-distortion performance. Code will be released at `https://github.com/qingshi9974/AuxT`.

## 1 Introduction

Recent advances in learned image compression (LIC) (Ballé et al., 2018; Minnen et al., 2018; Cheng et al., 2020; Liu et al., 2023; Li et al., 2024a) have been attracting increasing attention. LIC usually employs a pair of nonlinear analysis and synthesis transforms (Ballé et al., 2020) to achieve mapping between input images and their latent representations. The nonlinear transforms are jointly optimized along with uniform quantizer and entropy model to render superior rate-distortion (R-D) performance than traditional handcrafted image codecs.

Despite their promising R-D performance, existing LIC methods suffer from slow convergence on training. We take TCM (Liu et al., 2023), the recent state-of-the-art LIC model, as an example. TCM requires more than 2 million training iterations that take more than 15 days on an NVIDIA GeForce RTX 4090 GPU to train a model for a single R-D point, and needs 6 models at different rates to guarantee satisfactory R-D performance over the whole rate region. Moreover, low-efficiency training hinders fast fine-tuning for varying image datasets and downstream tasks or adaptation to variable target bitrates. This issue limits the practical application of LIC models.

In fact, efficient training of LIC models has not been thoroughly explored, despite the properties of nonlinear transforms have been studied. Previous studies (Duan et al., 2022; He et al., 2022; Li et al., 2024c) find that the analysis transform in LIC exhibits energy compaction property (*i.e.*, the energy of latent representation concentrates in several dominant channels), and further observe two characteristics of nonlinear transforms to realize energy compaction (*i.e.*, *feature decorrelation* and *uneven energy modulation*). However, they ignore to consider the evolution of energy compaction during training. In this paper, we further investigate this problem, and reveal that LIC models struggle to improve the efficiency of energy compaction through the training process. As

---

\*Corresponding authors: Wenrui Dai; Shaohui Li.

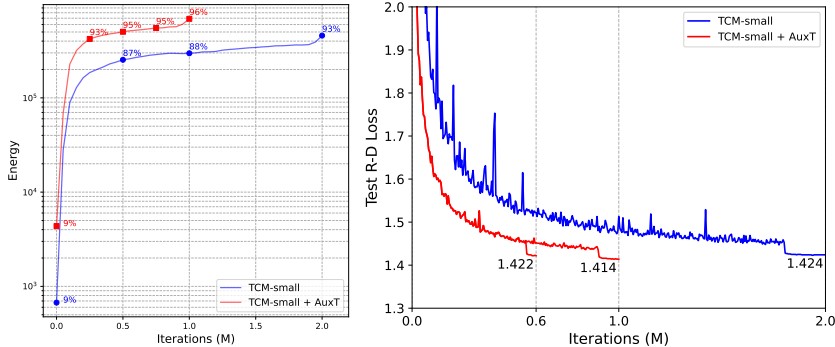

(a) Energy compaction vs. iterations    (b) Test R-D loss vs. iterations

Figure 1: Illustration on the evolution of energy compaction and R-D loss during training of TCM-small with and without the proposed auxiliary transform (AuxT). The energy is computed by the $L_2$ norm. (a) Energy compaction of the top 10% of channels with highest energy. The numbers on the curve is the corresponding energy ratio relative to the total energy of all channels. (b) Convergence curves of the R-D loss. Both TCM-small (Liu et al., 2023) and TCM-small+AuxT are trained with $\lambda$ set to 0.0483, and the energy and test R-D loss are averaged over the Kodak test set.

demonstrated in Figure 1a (blue line), during training, an LIC model could use only 10% channels of the latent representation to preserve 87% of its energy in the first 0.5 million iterations, but would require extra 1.5 million iterations to further increase the energy by 6%. This suggests the necessity to ease energy compaction in training nonlinear transform and thus improve training efficiency.

Motivated by these facts, we propose to disentangle the learning process of energy compaction for nonlinear transform with a simple yet effective linear auxiliary transform (AuxT). AuxT works as a bypass transform in addition to the nonlinear transform to achieve *coarse approximation* of feature decorrelation and uneven energy modulation. Such a design enables the nonlinear transform to focus on fitting the distribution of fine details and consequently yield faster convergence. Specifically, AuxT contains several wavelet-based linear shortcuts (WLSs) with each comprising a wavelet-based downsampling and an orthogonal linear projection for feature decorrelation and a subband-aware scaling for uneven energy modulation.

The auxiliary transform is lightweight and plug-and-play, which can be integerated seamlessly with existing LIC models. Figure 1 presents an instance of incorporating AuxT with TCM (specifically, the small version). Compared to vanilla TCM, TCM with AuxT rapidly reduces the channel-wise correlations of latent representations and evidently enhances energy compaction with identical training iterations. Consequently, the training process of LIC is significantly accelerated by introducing the lightweight AuxT.

To our best knowledge, this is one of the first successful attempt for LIC that *significantly accelerates* the convergence of training while achieving comparable or superior R-D performance. The contributions of this paper are summarized as below.

- We interpret the training process of LIC from the perspective of energy compaction property, and reveal the low efficiency of existing nonlinear transforms in achieving feature decorrelation and uneven energy modulation for energy compaction.
- We design a novel method to accelerate LIC training. Specifically, we propose a lightweight and plug-and-play auxiliary transform (AuxT) to achieve energy compaction with coarse feature decorrelation and uneven energy modulation.
- We demonstrate superior efficiency of AuxT on diverse LIC models, achieving approximately 40% to 70% reduction in training time with an average BD-rate reduction of 1.3%.

## 2 PRELIMINARY: LEARNED IMAGE COMPRESSION

Learned image compression models typically consist of two fundamental components: nonlinear transforms and an entropy model. The nonlinear transforms include a nonlinear analysis transform

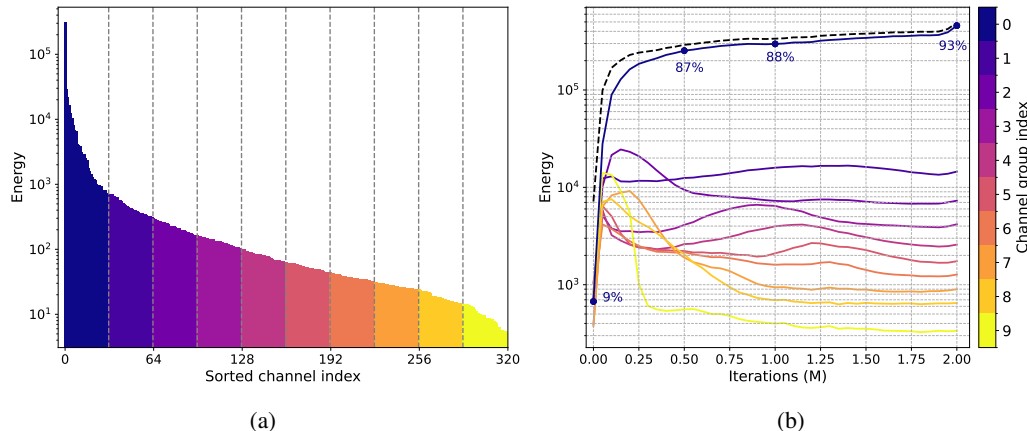

(a)                 (b)

Figure 2: Characteristics of energy distributions in latent representations for TCM-small (Liu et al., 2023), averaged over 24 images from the Kodak dataset. (a) Channel-wise energy distribution after convergence. We sort the channels of latent and split them into 10 groups. The energy is predominantly concentrated in the first group associated with low-frequency features, while other groups corresponding to high-frequency features carry significantly less energy. (b) Evolution of the energy distribution during the training process. The total energy of each group is plotted over the training process, with the dotted line representing the total energy of all channels. The numbers on the curve indicate the energy ratio of the group with the highest energy.

$g_a(\cdot; \boldsymbol{\theta}_a)$ with trainable parameters $\boldsymbol{\theta}_a$ at the encoder side and a nonlinear synthesis transform $g_s(\cdot; \boldsymbol{\theta}_s)$ with $\boldsymbol{\theta}_s$ at the decoder side. $g_a(\cdot)$ transforms the input image $\boldsymbol{x}$ into a latent representation $\boldsymbol{y} = g_a(\boldsymbol{x}; \boldsymbol{\theta_a})$, which is then discretized to $\hat{\boldsymbol{y}} = Q(\boldsymbol{y})$ using uniform quantization $Q(\cdot)$, whereas $g_s(\cdot)$ obtains the reconstructed image $\hat{\boldsymbol{x}} = g_s(\hat{\boldsymbol{y}}; \boldsymbol{\theta}_s)$ from the quantized latent $\hat{\boldsymbol{y}}$.

The probability of $\hat{\boldsymbol{y}}$ is usually estimated with multi-dimensional Gaussians with mean $\boldsymbol{\mu}$ and scale $\boldsymbol{\sigma}$, where the mean and scale are estimated by the entropy model using side information $\hat{\boldsymbol{z}}$ and contextual information $\phi$. The estimated probability is further employed in entropy coding that outputs a bitstream with an average length calculated by

$$\mathcal{R} = \mathbb{E}_{\boldsymbol{x} \sim p_x} \left[ -\log_2 p_{\hat{\boldsymbol{y}}}(\hat{\boldsymbol{y}}) \right] + \mathbb{E}_{\boldsymbol{x} \sim p_x} \left[ -\log_2 p_{\hat{\boldsymbol{z}}}(\hat{\boldsymbol{z}}) \right]. \tag{1}$$

Consequently, LIC models are optimized using the R-D loss, which incorporates a Lagrangian multiplier $\lambda$ that controls the trade-off between the rate $\mathcal{R}$ and the distortion $\mathcal{D}$, expressed as:

$$\mathcal{L}_{RD} = \mathcal{R} + \lambda \cdot \mathcal{D}. \tag{2}$$

According to Equation 2, different bitrates are realized using different values of $\lambda$. Please refer to Appendix A.1 for more related works on LIC.

## 3  ENERGY BASED INTERPRETATION OF LIC TRAINING

Understanding the training process of LIC is crucial to addressing the slow convergence issue. In this section, we begin by re-examining the results of previous work on the energy-compaction property of nonlinear transforms (Duan et al., 2022; He et al., 2022; Li et al., 2024c), and then show the two reasons why the energy-compaction property is hard to learn.

Traditional transform coding usually leverages orthogonal linear transforms such as Discrete Cosine Transform (DCT), Discrete Wavelet Transform (DWT), Karhunen-Loève Transform (KLT), and multiscale geometric analysis to aggregate most energy of the image into few components. And the energy-compaction property has a significant contribution to the compression performance. In an optimized LIC model, the majority of energy in the latent representation is concentrated in a few dominant channels, while the other channels contain considerably less energy (as illustrated in Figure 2). According to (Li et al., 2024c), the dominant channels holding high energy are low-frequency channels, whereas the others are high-frequency channels.

For a clear demonstration, we categorize these latent representation channels into 10 groups according to their average energy, and visualize the evolution of the energy ratio (*i.e.*, the ratio of energy in the total energy) for each group during the training process. Figure 2 shows that the LIC model is trained to gradually achieve energy compaction. Specifically, we obtain the following two observations.

*(i) High-energy channels exhibit a trend of increasing energy ratio, but the growth of both energy and energy ratio is slow.*

*(ii) Low-energy channels are not stable in energy ratios, and might adversely affect the training stability of LIC.*

These observations highlight the challenge of slow energy compaction in LIC training. To address this challenge, we further identify two concrete reasons behind it.

**Feature Decorrelation.** Traditional transform coding inherently obtains energy-compacting representation through orthogonal linear transforms that are well designed for decorrelating frequency components in a structured manner. LIC learns nonlinear transforms to adapt to the source distribution of natural images and also achieves decorrelation in latent representation. In Figure 3, we estimate the average similarity between each two channels based on their guided backpropagation patterns (Springenberg et al., 2015) on the Kodak dataset throughout the whole training process. The pairwise channel similarity suggests the correlation of any two channels of latent representations is gradually reduced during the training process. However, unlike traditional transform coding, LIC requires numerous iterations in training to achieve ideal decorrelation, since orthogonality is not guaranteed for the analysis transform.

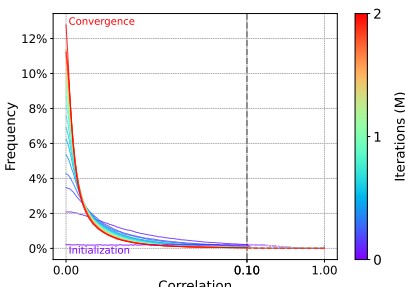

Figure 3: Normalized histograms of pairwise channel similarities in the analysis transform for different training iterations. We rescale the $x$-axis for better visualization. Please Zoom in for a better view

**Uneven Energy Modulation.** Traditional transform coding commonly applies different quantization steps on the transform coefficients of different frequencies. The low-frequency coefficients usually adopt smaller quantization steps than high-frequency coefficients to reduce average distortion. On the contrary, LIC models employ uniform quantizers with a fixed quantization step of 1 for all the latent representation. Low-frequency channels that inherently carry higher energy require additional amplitude amplification (*i.e.*, uneven energy modulation) to achieve finer quantization. Thus, the analysis transform gradually learns to amplify low-frequency channels, and overall energy in the latent representation increases as training proceeds. Figure 4 shows that the max intensity of latent representations is significantly scaled up from a value of 1 in the source space to over 100 in the latent space. However, the

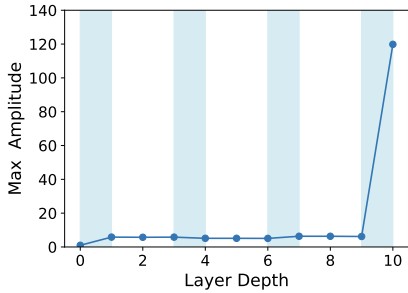

Figure 4: Maximum output intensity for each layer of the analysis transform $g_a$ of TCM-small (Liu et al., 2023). The blue area is subsampling layer.

scaling-up primarily occurs in the last layer of neural network realizing the analysis transform. This causes unbalanced gradients that could slow down the training process.

In light of these findings, we propose a method that disentangles the training process of the nonlinear transforms into feature decorrelation and uneven energy modulation to accelerate training.

## 4 AUXILIARY TRANSFORM FOR DISENTANGLED TRAINING

We introduce an auxiliary transform (AuxT) to disentangle the learning process of the nonlinear transform such that the difficulty in achieving energy compaction in training could be reduced and the convergence of training could be accelerated. The proposed AuxT works in parallel with

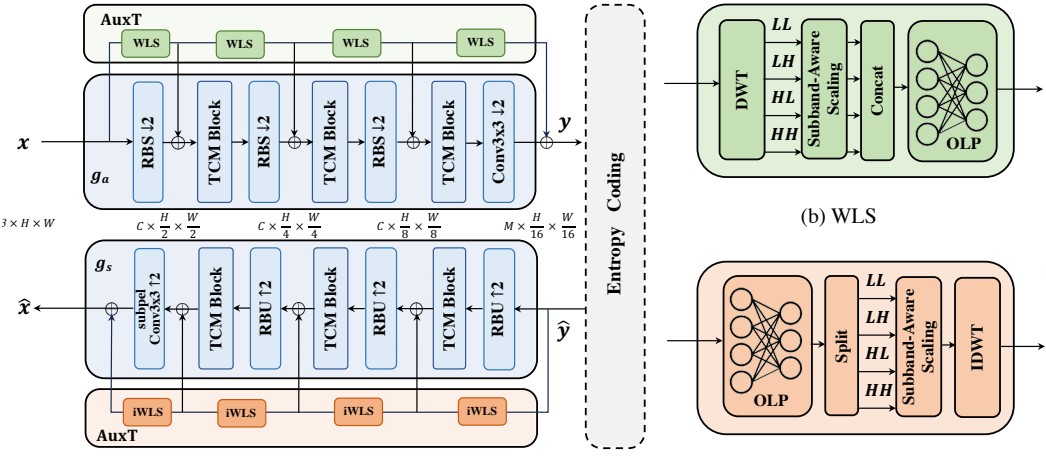

Figure 5: (a) Overview of the proposed method. Without loss of generality, we adopt the nonlinear transform from TCM (Liu et al., 2023) for illustration of our Auxiliary Transform (AuxT), while it can also integrate seamlessly with other LIC models. RBS and RBU denote the Residual Block with Stride and Residual Block Upsampling, respectively. Our method does not involve modifications to the entropy model, we omit the context model in this figure for simplicity. (b) Proposed wavelet-based linear shortrcut (WLS) for analysis transform, where DWT, subband-aware scaling and orthogonal linear projection (OLP) are performed sequentially. (c) iWLS for synthesis transform, which implements the inverse operation of WLS.

the nonlinear transform and efficiently provides a complementary latent representation with coarse decorrelation and energy modulation. This allows the nonlinear transform to pay more attention to learning latent representations for fine details, ultimately achieving higher compression performance with significantly reduced overall training effort. AuxT is achieved with multiple wavelet-based linear shortcuts (WLSs), where each WLS consists of three components in a sequence, *i.e.*, wavelet-based downsampling, subband-aware scaling, and orthogonal linear projection (OLP).

## 4.1 WAVELET-BASED LINEAR SHORTCUT (WLS)

**Wavelet-based down-sampling** is employed to achieve feature decorrelation. Besides, it achieves spatial down-sampling to align with the common design of LIC models. Let $P \in \mathbb{R}^{H \times W \times C}$ be the input of WLS, where $H$, $W$, and $C$ denote the height, width, and channel respectively. The input input $P$ is down-sampled by the Discrete Wavelet Transform (DWT). For simplicity, we use the Haar wavelet to decompose $P$ into four wavelet subbands $P_{LL} \in \mathbb{R}^{\frac{H}{2} \times \frac{W}{2} \times C}$, $P_{LH} \in \mathbb{R}^{\frac{H}{2} \times \frac{W}{2} \times C}$, $P_{HL} \in \mathbb{R}^{\frac{H}{2} \times \frac{W}{2} \times C}$, and $P_{HH} \in \mathbb{R}^{\frac{H}{2} \times \frac{W}{2} \times C}$. The filters to produce the four subbands are

$$f_{LL} = \frac{1}{2} \begin{bmatrix} 1 & 1 \\ 1 & 1 \end{bmatrix}, \; f_{LH} = \frac{1}{2} \begin{bmatrix} 1 & 1 \\ -1 & -1 \end{bmatrix}, \; f_{HL} = \frac{1}{2} \begin{bmatrix} 1 & -1 \\ 1 & -1 \end{bmatrix}, \; f_{HH} = \frac{1}{2} \begin{bmatrix} 1 & -1 \\ -1 & 1 \end{bmatrix}, \quad (3)$$

where $f_{LL}$ is the low-frequency filter, and $f_{LH}$, $f_{HL}$, and $f_{HH}$ are the high-frequency filters.

**Subband-aware scaling.** Considering lower-frequency channels require greater energy amplification (*i.e.*, uneven energy modulation) than higher-frequency channels, we design a subband-aware scaling method to adaptively amplify the energy of wavelet subbands. Specifically, each WLS employs four learnable scaling vectors, $s_{ll}, s_{lh}, s_{hl}, s_{hh} \in \mathbb{R}^C$ to scale the corresponding subbands $P_{LL}, P_{LH}, P_{HL}$, and $P_{HH}$, respectively. The scaled subbands are then concatenated along the channel dimension to form $\tilde{P} \in \mathbb{R}^{\frac{H}{2} \times \frac{W}{2} \times 4C}$ :

$$\tilde{P} = \text{Concat}(P_{ll} \odot e^{s_{ll}}, P_{lh} \odot e^{s_{lh}}, P_{hl} \odot e^{s_{hl}}, P_{hh} \odot e^{s_{hh}}), \quad (4)$$

where $\odot$ denotes channel-wise multiplication. The values of $s_{ll}, s_{lh}, s_{hl}, s_{hh}$ are initialized with 1, 0.5, 0.5, 0, respectively, reflecting uneven energy modulation for different subbands.

**Orthogonal Linear Projection (OLP).** Channel projection is considered to ensure that the output channels align with the intermediate features of the nonlinear transform. However, an unconstrained

linear projection layer is inferior in channel-wise decorrelation and causes excessive training time, due to the lack of orthogonality guarantees. To address this issue, we develop OLP by applying an orthogonality constraint $||\boldsymbol{W}^T\boldsymbol{W} - \boldsymbol{I}||_F^2$ to a $1 \times 1$ convolution layer $\mathbf{W} \in \mathbb{R}^{4C \times D}$. OLP projects $\tilde{\boldsymbol{P}}$ to obtain the final output of WLS, $i.e., \hat{\boldsymbol{P}} \in \mathbb{R}^{\frac{H}{2} \times \frac{W}{2} \times D}$. Please refer to Appendix A.5 for more details about the orthogonality constraint.

## 4.2 OVERALL ARCHITECTURE WITH MULTI-STAGE SHORTCUTS

As illustrated in Figure 5, we apply four stacked WLS in the analysis transform and four stacked iWLS in the synthesis transform. For the analysis transform, the output of each WLS will be added to the output of each down-sampling layer of the nonlinear transform $g_a$. The output of the last WLS (*i.e.*, $\hat{\boldsymbol{P}}_{final}$) will be added to the output of analysis transform $\boldsymbol{F}$ to form the final latent representation $\boldsymbol{y}$ that is subsequently quantized and encoded. For the synthesis transform, iWLS implements the inverse operation of WLS in the analysis transform, where the DWT is replaced by the inverse DWT (IDWT) and the channel-wise multiplication in Equation 4 is replaced by a channel-wise division to achieve corresponding energy reduction.

**Progressive Energy Modulation.** The orthogonality of the DWT and our OLP ensures that they act as energy-preserving transforms. In this structure, multiple subband-aware scaling operations in the multi-stage architecture can progressively and efficiently scales energy, which can avoid dramatic changes in energy and amplitude, leading to a more stable training process.

**Loss Function.** The proposed architecture is optimized in an end-to-end fashion using a combination of the R-D loss $\mathcal{L}_{RD}$ in Equation 2 and orthogonality regularization loss $\mathcal{L}_{orth}$ for OLP. The overall loss function is formulated as

$$\mathcal{L}_{overall} = \mathcal{L}_{RD} + \lambda_{orth}\mathcal{L}_{orth}, \text{ where } \mathcal{L}_{orth} = \sum_{\boldsymbol{W} \in \mathcal{W}} \left\| \boldsymbol{W}^\top \boldsymbol{W} - \boldsymbol{I} \right\|_F^2, \tag{5}$$

where $\mathcal{W}$ is the set of the weight matrix for all OLPs and $\lambda_{orth}$ is the regularization parameter.

## 5 EXPERIMENTS

### 5.1 EXPERIMENTAL SETUP

**Training.** We apply our AuxT to several mainstream LIC models to show its effectiveness, including mb2018mean (Minnen et al., 2018), ELIC (He et al., 2022), STF (Zou et al., 2022), and TCM (Liu et al., 2023). All the models are trained on the ImageNet-1k (Deng et al., 2009) dataset and optimized using Adam optimizer (Kingma & Ba, 2015). We set the batch size to 16 for convolution-based LIC models (Minnen et al., 2018; He et al., 2022) and 8 for transformer-based LIC models Zou et al. (2022); Liu et al. (2023). We train the models without our AuxT for 0.6M and 2M iterations respectively, and train the models with our AuxT for 0.6M and 1M iterations, respectively. The learning rate is initialized as $10^{-4}$ and is decayed by a factor of 10 after 0.55M iterations for 0.6M training iterations scenario, after 0.9M iterations for 1M training iterations scenario, and after 1.8M iterations for 2M training iterations scenario.

Two kinds of quality metrics, *i.e.*, mean square error (MSE) and multi-scale structural similarity (MS-SSIM), are used to measure the distortion $\mathcal{D}$. The Lagrangian multiplier $\lambda$ in the R-D loss used for training MSE-optimized models are $\{0.0025, 0.0035, 0.0067, 0.0130, 0.0250, 0.0483\}$, and those for MS-SSIM-optimized models are $\{2.40, 4.58, 8.73, 16.64, 31.73, 60.50\}$. The orthogonal regularization weight $\lambda_{orth}$ is 0.1. Experiments are performed on NVIDIA GeForce RTX 4090 GPU and Intel Xeon Platinum 8260 CPU.

**Evaluation.** We adopt three benchmark datasets, *i.e.*, Kodak image set (Kodak, 1993) with 24 images of $768 \times 512$ pixels, Tecnick testset (Asuni & Giachetti, 2014) with 100 images of $1200 \times 1200$ pixels, and CLIC Professional Validation dataset (CLIC, 2021) with 41 images of at most 2K resolution, for evaluations. We use both PSNR and MS-SSIM to measure the distortion, and bits per pixel (BPP) to evaluate bitrates. We present the PSNR-BPP results evaluated on Kodak in Table 1,and provide the results of other dataset and MS-SSIM metric in Appendix A.7.

Table 1: Performance comparison of the proposed method applied to various LIC anchor models on Kodak. We report the GMACs calculated using 768×512 input images for complexity comparison and GPU hours for training a single R-D point model for time comparison. BD-rate computed from PSNR-BPP curves by comparing with the anchor VTM-18.0 is adopted as the quantitative metric. The relative values represent the comparison with the anchor model trained for 2M iterations

| Model | # of Iterations | GMACs | | Training time | #Params | BD-rate |
|---|---|---|---|---|---|---|
| | (M) | Enc. | Dec. | (GPU hours) | (M) | (%) |
| *Convolution-based nonlinear transforms* | | | | | | |
| mbt2018mean (Minnen et al., 2018) | 0.6 | 44 | 43 | 15 | 17.6 | 34.3 |
| mbt2018mean (Minnen et al., 2018) | 2.0 | 44 | 43 | 50 | 17.6 | 25.5 |
| mbt2018mean + AuxT | 0.6 | 49 | 48 | 18 (-64%) | 18.6 | 26.4 (+0.9) |
| mbt2018mean + AuxT | 1.0 | 49 | 48 | 30 (-40%) | 18.6 | **22.6 (-2.9)** |
| ELIC (He et al., 2022) | 0.6 | 132 | 130 | 43 | 33.8 | -0.5 |
| ELIC (He et al., 2022) | 2.0 | 132 | 130 | 143 | 33.8 | -4.5 |
| ELIC + AuxT | 0.6 | 137 | 135 | 46 (-68%) | 34.8 | -4.0 (+0.5) |
| ELIC + AuxT | 1.0 | 137 | 135 | 76 (-47%) | 34.8 | **-5.7 (-1.2)** |
| *Transformer-based nonlinear transforms* | | | | | | |
| STF (Zou et al., 2022) | 0.6 | 143 | 161 | 35 | 99.8 | 4.6 |
| STF (Zou et al., 2022) | 2.0 | 143 | 161 | 116 | 99.8 | -3.2 |
| STF + AuxT | 0.6 | 144 | 162 | 36 (-68%) | 100.6 | -3.2 (-0) |
| STF + AuxT | 1.0 | 144 | 162 | 61 (-47%) | 100.6 | **-5.7 (-2.5)** |
| TCM-small (Liu et al., 2023) | 0.6 | 112 | 148 | 72 | 45.2 | -0.1 |
| TCM-small (Liu et al., 2023) | 2.0 | 112 | 148 | 240 | 45.2 | -5.3 |
| TCM-small + AuxT | 0.6 | 114 | 150 | 75 (-68%) | 45.8 | -4.8 (+0.5) |
| TCM-small + AuxT | 1.0 | 114 | 150 | 125 (-48%) | 45.8 | **-6.0 (-0.7)** |
| TCM-large (Liu et al., 2023) | 0.6 | 315 | 449 | 100 | 76.6 | -4.1 |
| TCM-large (Liu et al., 2023) | 2.0 | 315 | 449 | 330 | 76.6 | -10.6 |
| TCM-large + AuxT | 0.6 | 324 | 457 | 105 (-68%) | 78.2 | -9.7 (+0.9) |
| TCM-large + AuxT | 1.0 | 324 | 457 | 175 (-47%) | 78.2 | **-11.3 (-0.7)** |

## 5.2 EXPERIMENT RESULTS

Table 1 presents a thorough comparison of the proposed AuxT applied to diverse LIC models. Each anchor LIC model can achieve good R-D performance when trained for long iterations. However, when trained for only 0.6M iterations, these anchor models perform substantially worse due to their slow convergence. By integrating AuxT, we consistently achieve a significant acceleration on training. For instance, after 0.6M iterations, STF+AuxT achieves comparable BD-rate performance to STF trained for 2M iterations, reducing training time by 68%. Furthermore, training STF+AuxT for 1M iterations yields a 2.5% BD-rate improvement over the anchor model trained for 2M iterations, with only half the training time. Since AuxT is lightweight, it only bring a few additional parameter complexity (about 1M) and has little increase on inference complexity (1∼10 GMACs). In addition, Figure 1b shows the convergence curve for TCM-small (Liu et al., 2023), it clearly demonstrates that AuxT not only accelerates convergence but also enhances stability. We further compare the inference time in Appendix A.6 and provide the detailed R-D curves in Appendix A.8.

## 5.3 ANALYSIS ON FEATURE DECORRELATION AND ENERGY MODULATION

We analyze the energy and decorrelation behavior of our proposed auxiliary transform along with analysis transform to better understand the disentangling effect of proposed method.

**Energy distribution among channels of both branches.** Figure 6a illustrates the energy distribution among channels for both the nonlinear transform and AuxT when the model converges. We observe the following key points:

**(1)** The simple auxiliary transform can effectively capture the high-energy lower-frequency representations. In contrast, the complicated nonlinear transform focuses and contributes more on the intricate low-energy higher-frequency representations.

**(2)** By integrating AuxT, the nonlinear transform exhibits a more balanced energy distribution across channels, which facilitates a stable train when the architecture is sophisticated.

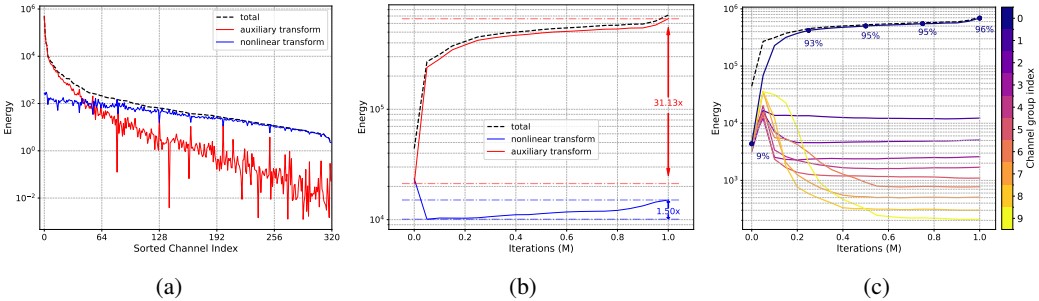

|  (a) | (b) | (c) |

Figure 6: Visualization of the energy distribution of our proposed auxiliary transform and the nonlinear transform. (a) Energy distribution among channels of different branches, with the dotted line indicating the total energy for both branches. (b) Evolution of energy during training of the two transforms, where the dotted line represents the total energy for both branches over time. (c) Evolution of energy distribution during training, channels are grouped same as Figure 2. The dotted line presents the total energy for all channels. Please Zoom in for a better view.

**Evolution of the energy of both branches.** Figure 6b shows the evolution of the total energy for both branches during training. AuxT and the nonlinear transform have similar energy at the beginning of the training process. As the training progresses, the energy of the auxiliary transform gradually increases, ultimately exhibiting a significant **31.1×** energy amplification. In contrast, the energy of the nonlinear transform exhibits a more stable trend, with only a sudden energy reduction in the early stages. After this drop, it only gradually exhibits a **1.5×** energy amplification. This phenomenon further demonstrates that uneven energy modulation is primarily achieved by AuxT, which alleviates the training burden on the nonlinear transform.

**Evolution of the energy of different Channels.** Figure 6c illustrates the evolution of total energy across different channels, where the grouping strategy is identical to Figure 2. We observe the following phenomena. **(1)** AuxT significantly enhances energy compaction. The top 10% high-energy channels account for **96%** of the total energy in the latent representation after 1M training iterations, compared to **93%** in the anchor model trained for 2M iterations (see Figure 2b). **(2)** Low-energy high-frequency channels exhibit a more stable energy evolution, showing almost no change in the latter half of training, compared to the unstable trend observed in the anchor model (see Figure 2b). Since these channels are primarily realized by the nonlinear transform, this further demonstrates that AuxT contributes to the stable training of the nonlinear transform.

**Decorrelation efficiency of both branches.** Figure 7 shows the pairwise channel similarity for both AuxT and the nonlinear transform. We observe that the nonlinear transform exhibits a strong decorrelation ability due to its sophisticated design, with most pairwise channel similarities being less than 0.01. Additionally, AuxT alone also demonstrates a decent level of decorrelation. When integrating both branches, the overall decorrelation is further enhanced, leading to even greater decorrelation efficiency and compression performance.

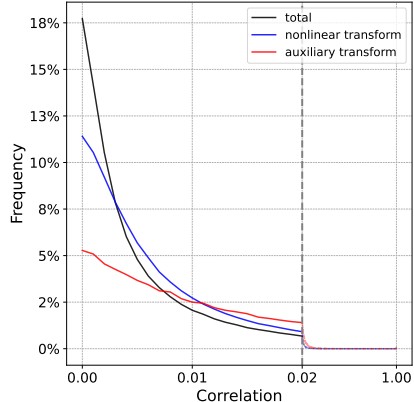

Figure 7: Normalized histograms of pairwise channel similarities for different branches. We rescale the $x$-axis for better visualization.

### 5.4 ABLATION STUDIES

We perform ablation studies to further evaluate the effectiveness of our proposed AuxT. Experiments are conducted on TCM-small (Liu et al., 2023), and each model is trained for 0.6M training iterations.

**Effect of the components of WLS.** We first evaluate the effectiveness of each WLS component through the following experiments: **[A1]** removing the orthogonal constraint for OLP; **[A2]** removing the subband-aware scaling; **[A3]** replacing DWT with an average pooling layer; and **[A4]** replacing DWT with a convolutional layer with a stride of 2. For **[A3]** and **[A4]**, we also remove the subband-aware scaling since DWT is no longer used.

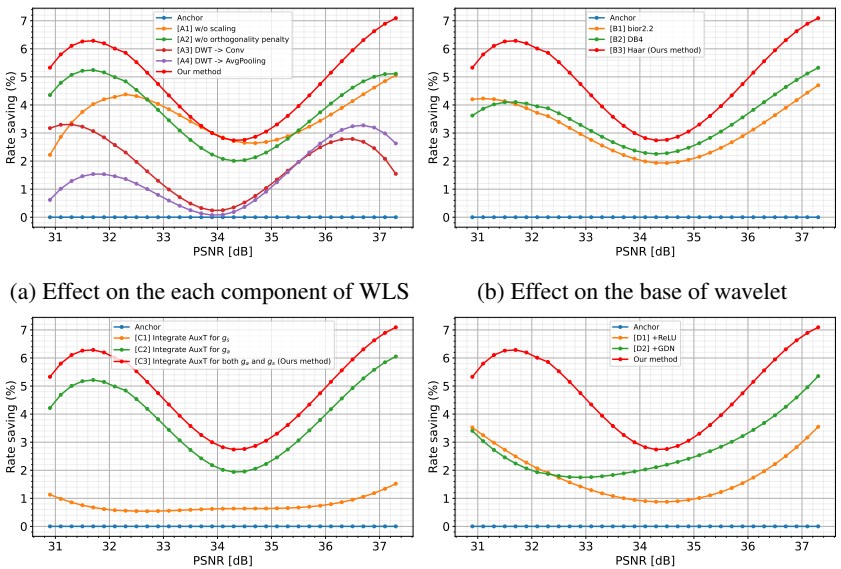

(a) Effect on the each component of WLS

(b) Effect on the base of wavelet

(c) Effect of AuxT in analysis and synthesis

(d) Effect of the linearity of AuxT

Figure 8: Ablation studies. We use the TCM-small as the anchor model and calculate the rate saving relatives to it. Please Zoom in for a better view.

As shown in Figure 8a, removing the subband-aware scaling results in a notable degradation in performance. However, replacing OLP or DWT with non-orthogonal layers leads to an even more significant performance drop, highlighting the importance of orthogonal linear transforms in WLS for stable energy modulation and enhanced feature decorrelation. Additionally, we observe that the frequency decomposition provided by DWT is essential and cannot be replaced by a simple low-frequency filter such as pooling.

**Effect of the bases of wavelet.** **[A3]** and **[A4]** highlight the importance of DWT in our WLS. We further evaluate the effect of different wavelet bases in DWT. As shown in Figure 8b, employing more complex wavelets like Daubechies wavelet **db4** and biorthogonal wavelet **bior2.2** leads to a performance degradation, indicating that the simple Haar wavelet is better.

**Effect of AuxT in analysis and synthesis.** We evaluate the effectiveness of AuxT in the analysis transform $g_a$ and the synthesis transform $g_s$. Figure 8c indicates that AuxT in $g_s$ yields trivial gain on convergence. The primary benefits come from implementing AuxT in $g_a$, highlighting the importance of enhancing the feature decorrelation and uneven energy modulation for $g_a$.

**Effect of the linearity of AuxT.** Unlike the nonlinear transform used in LIC, our AuxT is a purely linear transform without any nonlinear operators. We demonstrate the necessity of this linearity by introducing a nonlinear layer after the OLP. As shown in Figure 8d, adding either ReLU (Glorot et al., 2011) or GDN (Ballé et al., 2016) significantly hinders the performance of AuxT. This is because the nonlinearity causes energy attenuation and disrupts orthogonality, which negatively impacts effective energy compaction. This demonstrates that a simple linear module can be more effective for energy compaction. More ablation studies can be found in Appendix A.3.

## 6    CONCLUSION

In this paper, we propose AuxT, a lightweight and plug-and-play linear auxiliary transform that works as a bypass transform alongside the nonlinear transform to address the slow convergence issue for existing learned image compression (LIC) models. AuxT leverages multiple wavelet-based linear shortcuts to achieve coarse approximations for feature decorrelation and uneven energy modulation, allowing the nonlinear transform to learn simplified distributions, thus accelerating the training without sacrificing performance. Empirical results demonstrate that AuxT is effectively integrated into several mainstream LIC models and speeds up their training process by 2 times, with an average BD-rate reduction of 1%.

## ACKNOWLEDGEMENT

This work was supported in part by the National Natural Science Foundation of China under Grant 62320106003, Grant 62371288, Grant 62301299, Grant 62401357, Grant 62431017, Grant 62401366, Grant 62120106007, Grant 62125109, Grant U24A20251, and in part by the Program of Shanghai Science and Technology Innovation Project under Grant 24BC3200800.

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

# A APPENDIX

## A.1 RELATED WORK

**Learned Image Compression.** Learned image compression has seen significant advancements in recent years due to its impressive rate-distortion (R-D) performance. Unlike traditional transform coding methods that use linear transforms to decorrelate input images, LIC employs nonlinear transforms to achieve a more compact latent representation. Early works (Ballé et al., 2017; 2018; Minnen et al., 2018; Minnen & Singh, 2020) utilized convolutional neural networks (CNNs) to develop these nonlinear transforms, focusing on capturing local patterns in images. Later approaches incorporated non-local blocks (Cheng et al., 2020) to enhance the ability to capture global correlations. More recently, advancements have leveraged attention-based mechanisms and transformer architectures (Lu et al., 2022; Zou et al., 2022; Liu et al., 2023; Li et al., 2024a) for LIC, underscoring the idea that increasing complexity and capabilities can lead to more expressive nonlinear transforms.

Another key aspect of LIC research involves designing more effective entropy models. Earlier works (Ballé et al., 2017; 2018) introduced factorized and hyperprior models to boost compression efficiency. More recent efforts (Minnen et al., 2018; Minnen & Singh, 2020; Koyuncu et al., 2022; Qian et al., 2022) have incorporated spatial or channel-wise autoregression into entropy models, further enhancing their performance.

Recent works (Liu et al., 2022; 2024; Li et al., 2024b) in LIC have focused on developing image codec tailored for machine perception rather than solely for human perception, aiming to enhance the performance of downstream machine vision tasks while reducing storage and transmission costs.

Despite these advancements, many existing LIC methods suffer from long training times, requiring millions of iterations to converge effectively. This prolonged training duration poses significant challenges, hindering the real-world applications.

**Energy Compaction Property in LIC.** Energy compaction is a fundamental characteristic in traditional transform coding, indicating that most of the signal's energy is concentrated into a few components. In the field of LIC, several studies have explored this concept. For instance, Cheng et al. (2019b) and Cheng et al. (2019a) introduced a spatial energy compaction-based penalty in the loss function, encouraging energy concentration within a few channels. He et al. (2022) observed that the analysis transform in LIC inherently exhibits an information compaction property, where a limited number of channels carry significantly more average energy. Additionally, Li et al. (2024c) further demonstrated that higher-energy channels are typically characterized by low-frequency components, while lower-energy channels tend to carry high-frequency information. In our work, we find that the energy compaction process is closely linked to the slow training issue of LIC and propose an auxiliary transform to address this issue.

**Decorrelation for LIC.** Existing LIC methods often struggle with achieving better decorrelation ability to further remove the redundancies in the latent representation. The main solution is to leverage more complex neural network modules to enhance the overall representation ability of nonlinear transforms (Ballé et al., 2017; 2018; Minnen et al., 2018; Minnen & Singh, 2020; Lu et al., 2022; Zou et al., 2022; Liu et al., 2023; Li et al., 2024a). In addition, Ballé et al. (2016) proposed the Generalized Divisive Normalization (GDN) layer to helps to decorrelate information across channels. Guo et al. (2021) further explored the cross-channel relationships of the latents and achieve better channel-wise decorrelation by the proposed causal context prediction module.

**Training Acceleration for Neural Networks.** The acceleration of neural network training has been a heated topic in deep learning research.

One key approach to speeding up training is the development of advanced optimizers. In addition to widely-used adaptive learning rate methods like Adam (Kingma & Ba, 2015) and accelerated schemes such as Nesterov momentum Nesterov (1983), several recent approaches have emerged to further enhance training efficiency. For example, Goyal et al. (2017) introduced a scaling rule to adjust the learning rate, enabling faster training when using large mini-batches. Furthermore, You et al. (2017) and You et al. (2020) proposed layer-wise adaptive learning rates, which allow for the scaling of batch size, ultimately reducing training time.

Other research has focused on training neural networks in lower-dimensional subspaces and employing batch selection to improve efficiency. For example, Li et al. (2018) proposed the idea of training in a reduced random subspace to measure the intrinsic dimension of the loss objective. Later work by Gressmann et al. (2020) enhanced this approach by considering the layer-wise structure and re-drawing the random bases at each step, leading to improved training performance. Li et al. (2022) proposed a low-dimensional trajectory hypothesis, which extracts subspaces from historical training dynamics, significantly improving the dimensionality efficiency of neural network training. In addition, Hong et al. (2024) introduced Diversified Batch Selection (DivBS), a reference-model-free method that efficiently selects diverse and representative samples for machine learning training.

In the filed of learned image compression, a concurrent work (Anonymous, 2024) also focuses on training acceleration. This work is inspired by prior research on subspace training (Li et al., 2018; Gressmann et al., 2020; Li et al., 2022) and focuses on modeling the training dynamics of the parameters of LIC model. It reduces the training space dimension and decreases the number of active trainable parameters over time, thereby achieving lower training complexity for LIC models. However, Anonymous (2024) does not thoroughly investigate the underlying reasons of slow convergence in LIC models, nor does it incorporate the specific characteristics of LIC. In contrast, our method leverages these unique characteristics (*i.e.*, energy compaction and uneven energy modulation) and, for the first time, analyzes the training process of LIC from the perspective of energy compaction. We believe that our approach, in combination with the methods presented in Anonymous (2024), can complement each other and further accelerate convergence.

## A.2 DISCUSSION ON THE DWT IN LEARNED IMAGE COMPRESSION.

In end-to-end learned image compression, many works have introduced DWT into the nonlinear transform to enhance its decorrelation ability. Mishra et al. (2020) use DWT to preprocess the input image and separately encode its wavelet subbands. Ma et al. (2020) design a wavelet-like nonlinear transform using the lifting scheme. Fu et al. (2024) replace the down-sampling and up-sampling layers of LIC with DWT and IDWT, respectively, to better remove frequency-domain correlation. We highlight that our method differs from these methods as we do not modify the nonlinear transform itself, but instead introduce a bypass auxiliary transform to address the convergence issue. To further demonstrate the superiority of our method, we conduct experiments to test whether adopting DWT directly in the nonlinear transform can accelerate convergence. Specifically, we replace the sub-sampling and up-sampling layers of TCM with DWT and IDWT, respectively and append a Residual Block before and after the DWT and IDWT layers to maintain the parameter count.

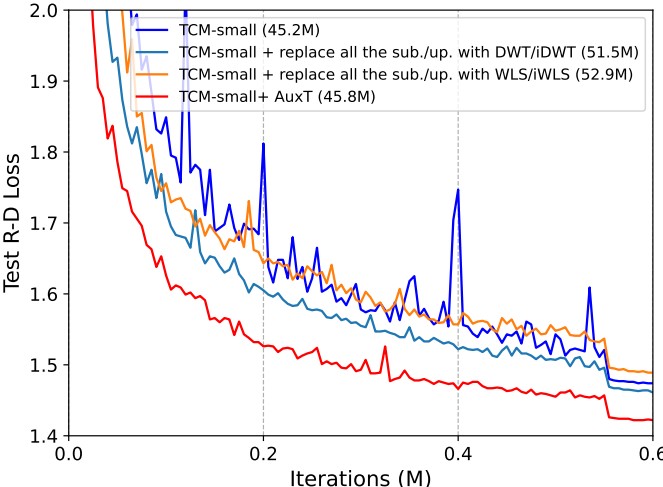

Figure 9: Convergence curves of the test R-D loss. $\lambda$ is set as 0.0483.

Figure 9 shows the convergence curves of different methods. We observe that replacing all the sub-sampling and up-sampling layers with wavelet yields a slight performance gain compared to

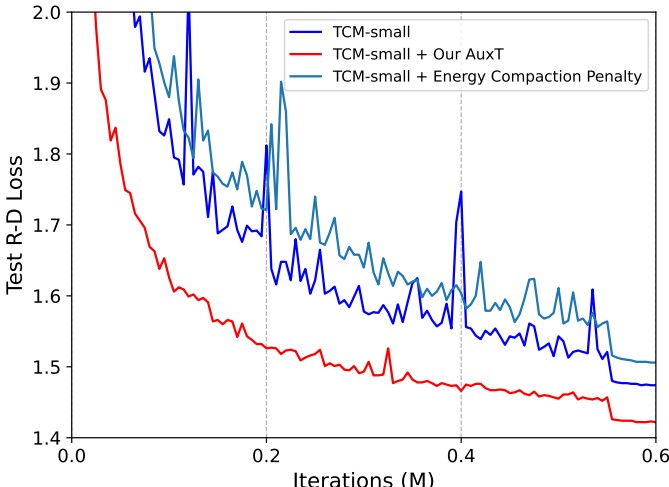

Figure 10: Convergence curves of the test R-D loss. $\lambda$ is set as 0.0483.

Table 2: Effect of the orthogonality regularization weight ($\lambda_{orth}$) on the R-D performance. The TCM-small+AuxT model is trained for 0.6M iterations by $\lambda$ set as 0.0483. and the R-D losses are evaluated on the Kodak dataset.

| $\lambda_{orth}$ | Rate (bpp) | Distortion (PSNR) | Test R-D Loss |
|---|---|---|---|
| 0 | 0.8958 | 37.726 | 1.447 |
| 0.01 | 0.8890 | 37.816 | 1.429 |
| **0.1** | 0.8865 | 37.834 | **1.424** |
| 1.0 | 0.8950 | 37.742 | 1.445 |

anchor model, but it also results in higher parameter complexity (51.5M vs. the original 45.2M). In addition, replacing all the sub-sampling and up-sampling layers with proposed WLS and iWLS leads to a performance drop. This may be because these additional modules (subband-aware scaling, OLP) and orthognoal constraints hinder the nonlinear transforms from effectively learning latent representations for fine details. This further highlights the importance of the additional parallel branch design in accelerating convergence.

### A.3 ADDITIONAL ABLATION STUDIES

**Compared with Energy Compaction Penalty.** Cheng et al. (2019b) and Cheng et al. (2019a) introduce a spatial energy compaction-based penalty to encourage the nonlinear transform of Learned Image Compression (LIC) to exhibit strong energy concentration properties. We also apply this penalty in the TCM-small (Liu et al., 2023), with the convergence curve shown in Figure 10. Our observations indicate that this spatial energy compaction-based penalty does enhance energy compaction, with the top 10% of energy channels accounting for 99% of the total energy, compared to 96% for TCM-small + AuxT and 90% for TCM-small. However, this strong penalty leads to a degradation in rate-distortion (R-D) performance and the training progress is still unstable. In contrast, our AuxT enhances energy compaction in a structured manner, which benefits both convergence and overall R-D performance.

**Effect of $\lambda_{orth}$.** We further evaluate the effect of the weight of orthogonality regularization loss. We train the TCM-small+AuxT with different value of $\lambda_{orth}$ for 0.6M iterations and the R-D losses evaluated on Kodak dataset are listed in the Table 2. The results indicate the the best choice of $\lambda_{orth}$ is 0.1.

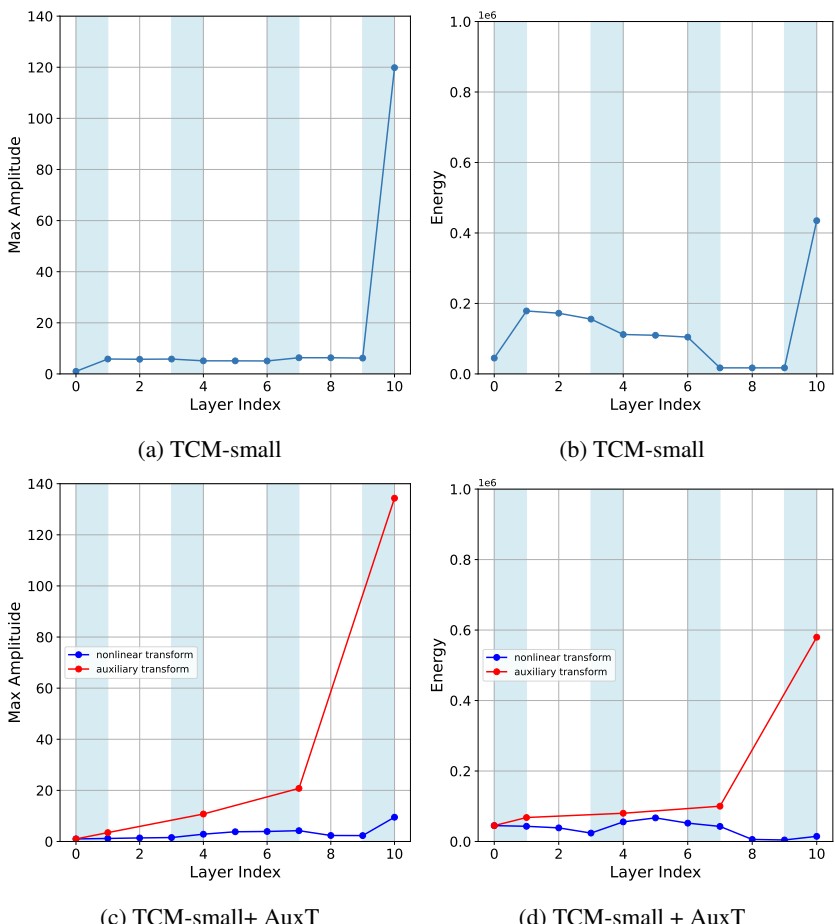

Figure 11: (a) Maximum output intensity for each layer of the analysis transform $g_a$ of TCM-small (b)Average energy for each layer of the analysis transform $g_a$ of TCM-small (c) Maximum output intensity for each layer of the analysis transform $g_a$ and auxiliary transform of TCM-small+Aux (d) Average energy for each layer of the analysis transform $g_a$ and auxiliary transform of TCM-small+Aux.

### A.4 ENERGY MODULATION BEHAVIOR

Figure 11 shows the effect of our subband-aware scaling which can progressively achieve energy modulation, thus avoiding dramatic changes in energy and amplitude, leading to a more stable training prcocss.

### A.5 DETAILS FOR ORTHOGONAL CONSTRAINT.

The orthogonality concept has been widely explored in the filed of deep learning (Le et al., 2011; Vorontsov et al., 2017; Bansal et al., 2018; Wang et al., 2020), as it implies energy preservation and encourages filter diversity. In our work, we use the orthogonal kernel regularization to force our orthogonal linear projection $W \in \mathbb{R}^{4C \times D}$ as orthogonal layer. Specifically, if $4C > D$, $W$ is an undercomplete matrix, and we use a row orthogonal regularizer $||W^T W - I||_F^2$, where $I \in \mathbb{R}^{D \times D}$ is the identify matrix. Conversely, if $4C < D$, $W$ becomes an overcomplete matrix, and column orthogonality is encouraged using the penalty $||WW^T - I'||_F^2$. In our practical implementation, we use the row orthogonal regularizer for both cases, since the row orthogonality and column orthogonality are equivalent in the mean squared error (MSE).

**Lemma A.1** *The row orthogonality and column orthogonality are equivalent in the mean squared error (MSE), i.e., $||W^T W - I||_F^2 = ||WW^T - I'||_F^2 + U$, where $U$ is a constant.*

Table 3: Comparison on the coding complexity of the proposed method applied to various LIC anchor models. The entropy coding time is excluded.

| Model | Enc. Inference Time (ms) | Dec. Inference Time (ms) |
|---|---|---|
| *Convolution-based nonlinear transforms* | | |
| mbt2018mean | 35.3 | 11.8 |
| mbt2018mean + AuxT | 35.4 | 13.2 |
| ELIC | 74.2 | 53.5 |
| ELIC + AuxT | 76.7 | 55.6 |
| *Transformer-based nonlinear transforms* | | |
| STF | 104.4 | 110.3 |
| STF + AuxT | 106.0 | 111.6 |
| TCM-small | 185.2 | 183.8 |
| TCM-smal + AuxT | 188.6 | 186.4 |
| TCM-large | 216.4 | 215.6 |
| TCM-large + AuxT | 222.1 | 219.6 |

The following proof is provided in the supplementary material of Le et al. (2011). We would like to present it here for the reader's convenience.

*Proof.* For $\boldsymbol{W} \in \mathbb{R}^{M \times N}$ as an arbitrary matrix, we denote $\|\boldsymbol{W}^T\boldsymbol{W} - \boldsymbol{I}_N\|_F^2$ as $L_r$ and $\|\boldsymbol{W}\boldsymbol{W}^T - \boldsymbol{I}_M\|_F^2$ as $L_c$.

$$
\begin{aligned}
L_r =& \|\boldsymbol{W}^T\boldsymbol{W} - \boldsymbol{I}_N\|_F^2 \\
=& \operatorname{tr}\left[(\boldsymbol{W}^T\boldsymbol{W} - \boldsymbol{I}_N)^T(\boldsymbol{W}^T\boldsymbol{W} - \boldsymbol{I}_N)\right] \\
=& \operatorname{tr}(\boldsymbol{W}^T\boldsymbol{W}\boldsymbol{W}^T\boldsymbol{W}) - 2\operatorname{tr}(\boldsymbol{W}^T\boldsymbol{W}) + \operatorname{tr}(\boldsymbol{I}_N) \\
=& \operatorname{tr}(\boldsymbol{W}\boldsymbol{W}^T\boldsymbol{W}\boldsymbol{W}^T) - 2\operatorname{tr}(\boldsymbol{W}\boldsymbol{W}^T) + \operatorname{tr}(\boldsymbol{I}_M) + N - M \\
=& \operatorname{tr}\left[\boldsymbol{W}\boldsymbol{W}^T\boldsymbol{W}\boldsymbol{W}^T - 2\boldsymbol{W}\boldsymbol{W}^T + \boldsymbol{I}_M\right] + N - M \\
=& \operatorname{tr}\left[(\boldsymbol{W}\boldsymbol{W}^T - \boldsymbol{I}_M)(\boldsymbol{W}\boldsymbol{W}^T - \boldsymbol{I}_M)\right] + N - M \\
=& \|\boldsymbol{W}\boldsymbol{W}^T - \boldsymbol{I}_M\|_F^2 + N - M \\
=& L_c + U
\end{aligned}
\tag{6}
$$

where $U = N - M$.

### A.6 COMPARISON ON THE CODING COMPLEXITY

We compare the coding complexity of the proposed AuxT the proposed method applied to various LIC anchor models. The coding complexity is measured by inference latency during encoding and decoding process, where the entropy coding time is excluded. The experiments show that the increase on the additional latency time caused by our AuxT can be ignored.

### A.7 ADDITIONAL BD-RATE RESULTS

Table 4 provide the BD-rate results for MS-SSIM metric evaluated on Kodak dataset (Kodak, 1993) and Table 5 provide the BD-rate results for PSNR metric evaluated on the CLIC (CLIC, 2021) and Tecnick (Asuni & Giachetti, 2014) dataset.

### A.8 ADDITIONAL R-D CURVES.

As shown in Figure 12, Figure 13, Figure 14, Figure 15, and Figure 16, we present the detailed rate-distortion (R-D) curves obtained by integrating our proposed auxiliary transform (AuxT) with various LIC anchor models at different training iterations.

Table 4: Comparison on the performance of the proposed method applied to TCM (Liu et al., 2023). The BD-rate is computed from MS-SSIM-BPP curves evaluated on the Kodak (Kodak, 1993) dataset as the quantitative metric with VTM-18.0 as the anchor.

| Model | # of Iterations (M) | BD-rate (%) with MS-SSIM Kodak (Kodak, 1993) |
|---|---|---|
| *Convolution-based nonlinear transforms* | | |
| TCM-small | 2.0 | -44.6 |
| TCM-small + AuxT | 1.0 | **-45.7 (-1.1)** |
| TCM-large | 2.0 | -48.3 |
| TCM-large + AuxT | 1.0 | **-49.0 (-0.7)** |

Table 5: Comparison on the performance of the proposed method applied to various LIC anchor models. The BD-rate is computed from PSNR-BPP curves evaluated on the CLIC (CLIC, 2021) and Tecnick (Asuni & Giachetti, 2014) dataset as the quantitative metric with VTM-18.0 as the anchor.

| Model | # of Iterations (M) | BD-rate (%) with PSNR Tecnick (Asuni & Giachetti, 2014) | CLIC (CLIC, 2021) |
|---|---|---|---|
| *Convolution-based nonlinear transforms* | | | |
| mbt2018mean | 2.0 | 21.9 | 26.4 |
| mbt2018mean + AuxT | 1.0 | **20.6 (-1.3)** | **24.2(-2.2)** |
| ELIC | 2.0 | -7.6 | -2.0 |
| ELIC + AuxT | 1.0 | **-8.2 (-0.6)** | **-3.2 (-1.2)** |
| *Transformer-based nonlinear transforms* | | | |
| STF | 2.0 | -5.6 | -1.6 |
| STF + AuxT | 1.0 | **-6.6 (-1.0)** | **-3.7 (-2.1)** |
| TCM-small | 2.0 | -5.9 | -3.2 |
| TCM-small + AuxT | 1.0 | **-6.6 (-0.7)** | **-4.0 (-0.8)** |
| TCM-large | 2.0 | -11.4 | -8.0 |
| TCM-large + AuxT | 1.0 | **-11.9 (-0.5)** | **-8.4 (-0.4)** |

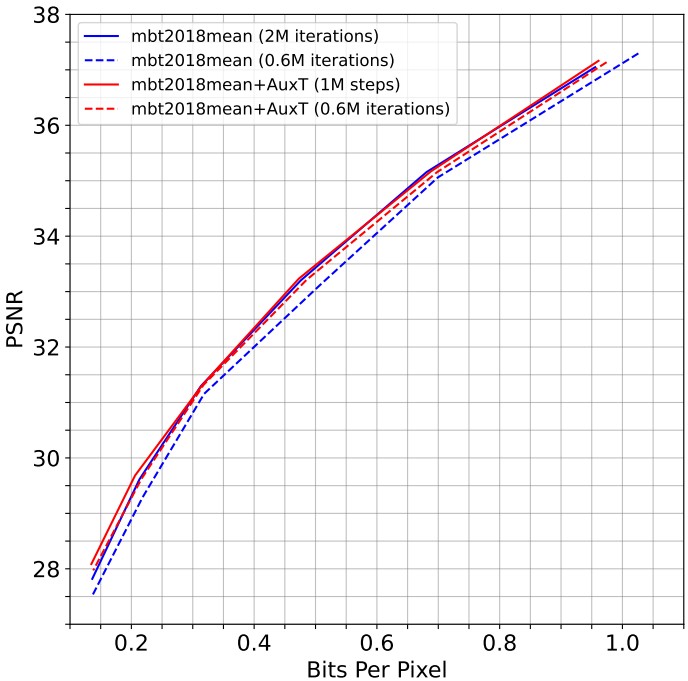

Figure 12: R-D curve for mbt2018mean

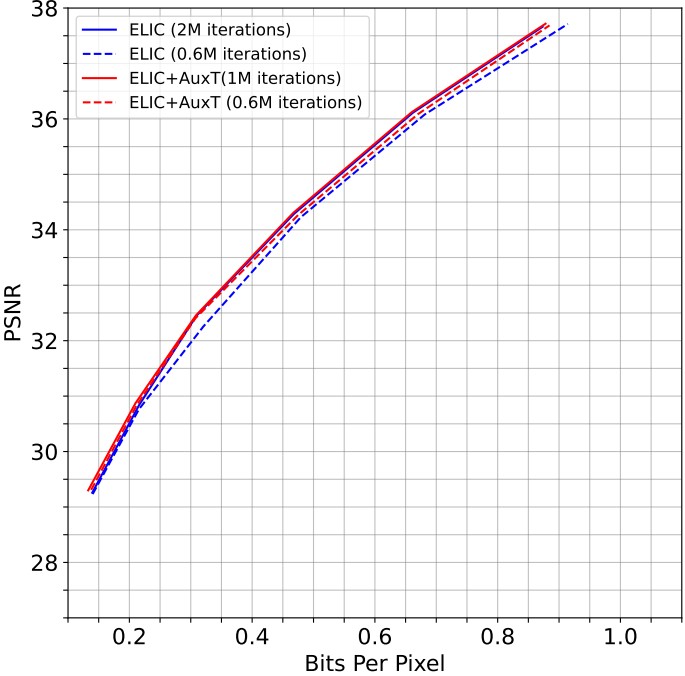

Figure 13: R-D curve for ELIC

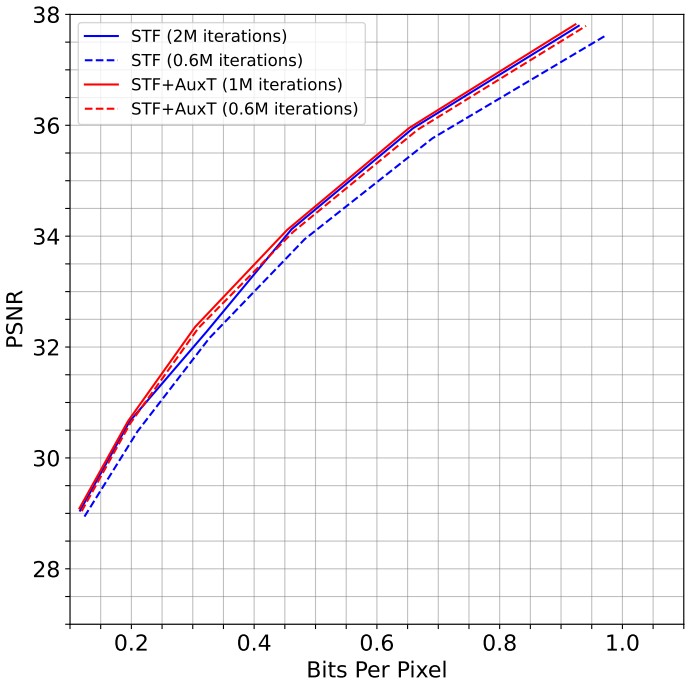

Figure 14: R-D curve for STF

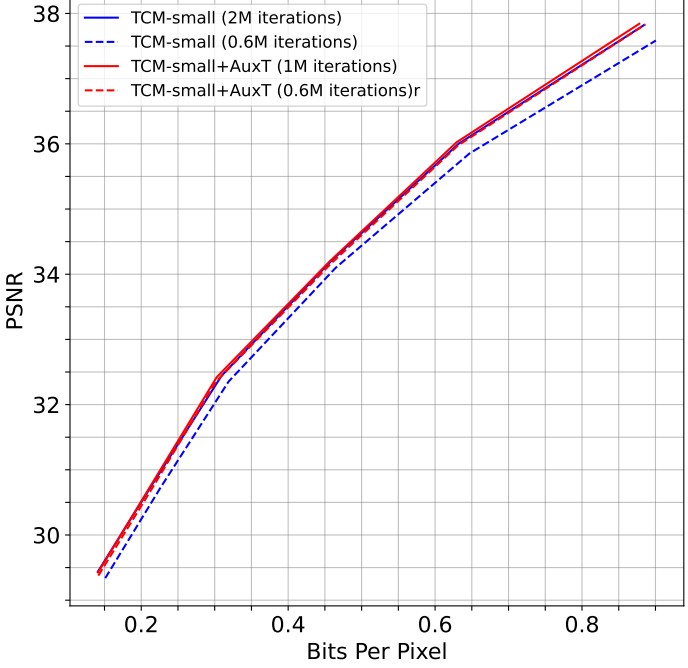

Figure 15: R-D curve for TCM-small

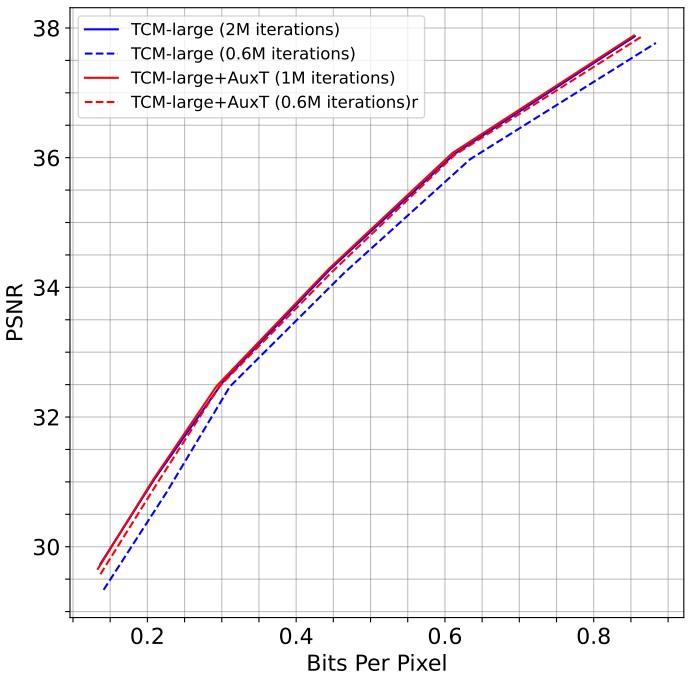

Figure 16: R-D curve for TCM-large

