# OpenReview forum: "On Disentangled Training for Nonlinear Transform in Learned Image Compression"
_ICLR.cc/2025/Conference — ICLR 2025 Spotlight_

### Official Review · Reviewer_faRP · 2024-10-28

**Soundness:** 3
**Presentation:** 3
**Contribution:** 4
**Rating:** 8
**Confidence:** 4

**Summary:**

Overall, this paper presents a linear auxiliary transform (AuxT) to disentangle energy compaction in training nonlinear transforms. By analyzing the characteristics of the LIC framework, the authors find the factors that caused the slow convergence of the LIC. Their methods accelerate the training of LIC models by around 2 times with compatible performance.

**Strengths:**

1. The authors analyze the training process of LIC from the perspective of energy compaction and highlight the inefficiency of existing nonlinear transforms in achieving effective feature decorrelation and uneven energy modulation necessary for optimal energy compaction.

2. The authors propose an auxiliary transform (AuxT) specifically designed to enhance energy compaction by enabling coarse feature decorrelation and uneven energy modulation.

3. Experimental results demonstrate that their method can reduce training time by approximately 45%.

Overall, this is an interesting and insightful paper. The authors propose a straightforward method that results in significant improvements.

**Weaknesses:**

1. Typos: For example, in the sentence: "and further observe two characteristics of nonlinear transforms to realize energy (i.e., feature decorrelation and uneven energy modulation)," there seems to be a missing word. It would be clearer to say "realize energy compaction" to fully capture the intended meaning.

2. Phrasing Recommendation: (1). Instead of using the phrase "this is the first successful attempt", which may not be entirely accurate, consider revising the sentence to:

"To the best of our knowledge, this is one of the first successful attempts in LIC that significantly accelerates the convergence of training while achieving comparable or superior R-D performance."

Additionally, it would be beneficial to include the earlier contemporaneous work that also explored training acceleration for LIC in the RELATED WORK section. You could reference:
"Accelerating Learned Image Compression with Sensitivity-aware Embedding and Moving Average" (https://openreview.net/forum?id=V1sChEsXZg).

(2). "Experimental results demonstrate that the proposed approach can accelerate the training of LIC models by 2 to 3 times while simultaneously achieving an average 1% BD-rate reduction."

However, based on Table 1, when the training time is reduced to approximately one-third of the original, there is no observed BD-rate reduction. I suggest revising the statement to: "can accelerate the training of LIC models by 2 times while simultaneously achieving an average 1% BD-rate reduction."


3. It would be important to add a subsection in RELATED WORK discussing the acceleration of training in other tasks. This could provide a broader context and demonstrate awareness of related methodologies in the field. You could discuss approaches from other domains that have addressed similar challenges in accelerating the training process.

**Questions:**

Please refer to the weaknesses section. I am open to increasing my rating if the authors further enhance the quality of their manuscript.

---

> ### Author Response · Authors · 2024-11-22
> **Response to Reviewer faRP**
>
> **W1 : Typos and Phrasing Recommendation**
> Thanks for your valuable suggestion. We have modified our manuscript accordingly to further enhance the quality of the manuscript.
>
> **W2 : Discussion with the contemporaneous work [R8]**
> Thank you for providing this important contemporaneous work [R8], which is currently under review for TMLR.  We have discussed  [R8] in the RELATED WORK section in our revision. Our method differs from this work in both motivation and methodology. The core of this paper focuses on  modelling the training dynamics of the parameters, reducing the training space dimension, and decreasing the number of active trainable parameters over time, thereby achieving lower training complexity.  We notice that while this paper provides valuable insights, it does not deeply explore the underlying reasons for the slow convergence of LIC or incorporate the specific characteristics of LIC. In contrast, our method leverages these characteristics  (*i.e.*, energy compaction and uneven energy modulation). We make the first attempt to analyze the training process of LIC from the perspective of energy compaction and reveal the training challenges in the feature
> decorrelation and uneven energy modulation. I believe our approach, in combination with the methods in this paper, can complement each other and further accelerate convergence.
>
> **W3 : Related work for training acceleration**
> Thank you for your valuable advice. We have added a subsection in the RELATED WORK section discussing the acceleration of training in other tasks. This will provide a broader context and demonstrate our awareness of related methodologies in the field.
>
> [R8] Anonymous. Accelerating learned image compression with sensitivity-aware embedding and moving average. Submitted to Transactions on Machine Learning Research, 2024. URL https://openreview.net/forum?id=V1sChEsXZg. Under review

---

> > ### Comment · Reviewer_faRP · 2024-11-22
> > **Feedback to authors**
> >
> > Thank you for addressing my concerns in the revised manuscript. I appreciate the effort you have put into improving the quality and clarity of the paper. I will increase my rating accordingly. I look forward to seeing the impact of this work on the community.

---

> > > ### Author Response · Authors · 2024-11-23
> > >
> > > Thank you for your active feedback and engagement during the rebuttal process. We appreciate your endorsement!

---

### Official Review · Reviewer_5Rti · 2024-10-28

**Soundness:** 3
**Presentation:** 3
**Contribution:** 2
**Rating:** 6
**Confidence:** 5

**Summary:**

The aim of this paper is to accelerate the convergence of learned image compression models. The authors observe the phenomena of energy compaction and decorrelation in learned image compression training. A portion of the channels in the model takes up most of the energy, and the energy compaction becomes more pronounced as the number of iteration steps increases. For a learned model, it usually requires many iterations to achieve energy compaction. This paper proposes the use of a wavelet transform based bypass transform module, to achieve coarse approximation of feature decorrelation and uneven energy modulation, which enhances the convergence speed of the model.

**Strengths:**

1) The experimental results demonstrate the effectiveness of the bypassed wavelet transform module, and the model with the wavelet transform module requires only 1M iterations to achieve comparable rate-distortion performance compared to the original model with 2M iterations.

2) The authors demonstrate the validity of their approach on most of the existing models (e.g. ELIC, Minnen’18, STF, TCM).

**Weaknesses:**

1) The introduction of AuxT will introduce additional hyperparameters ($\lambda_{orth}$), will the optimal $\lambda_{orth}$ be different for different models? According to Table 2 in the Supplementary Material, different $\lambda_{orth}$ will have a large impact on the performance, how to choose the optimal $\lambda_{orth}$?

2) In the training of image compression models, is the difficulty in aggregating energy due to the presence of entropy constraints? In the presence of entropy constraints, in order to avoid large bit rates, the model would automatically reduce the rate of information aggregation to match the entropy model's ability to estimate redundancy? In fact, in the training of the DCVC series [1], only reconstruction loss training would be performed first, and then entropy constraints would be introduced later, and I intuitively think that this approach would accelerate the training of the model. Please compare with this approach (e.g. performing 200,000 reconstruction error training followed by 400000 joint training).

3) Also, there are currently methods that will train a small model (e.g. only hyperprior) [2] and then reload the trained transform module and then add a more complex entropy model, in terms of total time, how does this compare to the training time of the method that introduces AuxT (e.g. TCM & ELIC)?

References

[1] $\mathcal{L}\_{reconstruction}$ and $\mathcal{L}\_{contextual coding}$ for inter compression model in Table 4 in https://arxiv.org/pdf/2109.15047

[2] Sec B.1 TRAINING in https://openreview.net/pdf?id=IDwN6xjHnK8

**Questions:**

Please refer to weakness.

---

> ### Author Response · Authors · 2024-11-22
> **Response to Reviewer 5Rti**
>
> ---
>
> **Highlighting our contributions**
>
> We emphasize that our key contribution lies in being the first to analyze the convergence issue of LIC from the perspective of energy compaction and introducing an auxiliary bypass transform to address this issue. It is important to note that our approach differs significantly from existing works that simply modify the loss function for progressive training [R6] or adopt multi-stage training [R7]. Instead, we delve deeply into the characteristics of LIC and propose a novel architecture specifically designed to tackle the convergence challenge and successfully achieve 2$\times$ acceleration with 1\% BD-rate reduction. We believe our analysis and exploration of the convergence issue in LIC can shed light on future research direction of LIC.
>
>
> ---
>
> **W1: How to choose the optimal $\lambda_{orth}$**.
>
> Thank you for your suggestion. We selected the optimal value of $\lambda_{orth} = 0.1$ based on experimental results on TCM in our previously submitted manuscript and directly applied it to other models. We further validate the impact of different $\lambda_{orth}$ values on ELIC and mbt2018-mean. As shown in the table below, $\lambda_{orth} = 0.1$  is demonstrated to consistently obtain the best performance. This indicates that the choice of $\lambda_{orth}$ is not sensitive to the model and can be adapted to different networks.
>
> *Table R6: Effect on the $\lambda_{orth}$ for ELIC. Models are trained for 600k iterations with $\lambda=0.0483$.*
> | **ELIC + AuxT**            | **Distortion(PSNR)** | **Rate(bpp)** | **Test R-D Loss** |
> |----------------------------|----------------------|---------------|-------------------|
> | $\lambda_{orth}=0$         | 37.782               | 0.9115        | 1.4550            |
> | $\lambda_{orth}=0.01$      | 37.808               | 0.9082        | 1.4485            |
> | **$\lambda_{orth}=0.1$**       | **37.687**               | **0.8845**       | **1.4421**            |
> | $\lambda_{orth}=1$         | 37.787               | 0.9101        | 1.4524            |
>
> *Table R7: Effect on the $\lambda_{orth}$ for mbt2018mean. Models are trained for 600k iterations with $\lambda=0.0483$.*
> | **mbt2018mean + AuxT**     | **Distortion(PSNR)** | **Rate(bpp)** | **Test R-D Loss** |
> |----------------------------|----------------------|---------------|-------------------|
> | $\lambda_{orth}=0$         | 37.134               | 0.9819        | 1.6105            |
> | $\lambda_{orth}=0.01$      | 37.142               | 0.9785        | 1.6071            |
> | **$\lambda_{orth}=0.1$**       | **37.138**               | **0.9742**        | **1.6026**            |
> | $\lambda_{orth}=1$         | 37.187               | 0.9942        | 1.6177            |
>
> **W2: Effect of entropy constraints on energy aggregation**
>
> In fact, the presence of entropy constraints **does promote** more significant energy aggregation (*i.e.,* energy compaction) rather than reducing the rate of aggregation. This is because entropy is lower when energy is concentrated in fewer channels, while the entropy is maximized when the energy is evenly distributed across all channels. In this way, entropy constraints can force the model to concentrate the information in low-frequency channels.
>
> We have conducted experiments to further verify it. Without entropy constraints, the top 10% high-energy channels only account for 26.1% of the total latent energy after 300k iterations. However, with entropy constraints, they account for  66.3% after 300k iterations, while our method with entropy constraints achieves 94.4%. This demonstrates significantly more efficient energy aggregation.
>
> *Table R8: Energy ratio of top 10% channels with highest energy at different training stages.*
> | **Model**                                  | **Energy ratio (100K iterations)** | **Energy ratio (200K iterations)** | **Energy ratio (300K iterations)** |
> |--------------------------------------------|------------------------------|------------------------------|------------------------------|
> | TCM-small (w/o entropy constraints)       | 22.9%                        | 24.6%                        | 26.1%                        |
> | TCM-small (w/ entropy constraints)        | 56.1%                        | 62.2%                        | 66.3%                        |
> | TCM-small + AuxT (w/ entropy constraints) | 91.8%                        | 93.4%                        | 94.4%                        |

---

> ### Author Response · Authors · 2024-11-22
> **Response to Reviewer 5Rti （Part 2）**
>
> **W2: Effect of entropy constraints on energy aggregation (continue)**
>
> We also have followed your suggestion to perform the progressive training strategy for inter-frame compression of DCVC [R6] (i.e., performing 200,000 reconstruction error training iterations followed by 400,000 joint training iterations), denoted as *progressive training*. The R-D performance results of TCM-small and ELIC are shown in the table below.
>
> *Table R9: Comparison with the progressive training of DCVC [R6] for ELIC. All the models are trained for total 600k iterations with $\lambda=0.0483$.*
> | **Model (at 600k iterations)** | **Distortion(PSNR)** | **Rate(bpp)** | **Test R-D Loss** |
> |---------------------------|----------------------|---------------|-------------------|
> | ELIC (anchor model)        | 37.713               | 0.9143        | 1.4660            |
> | ELIC + progressive training| 37.711               | 0.9218        | 1.4736            |
> | **ELIC + AuxT**                | **37.687**               | **0.8845**        | **1.4421**            |
>
> *Table R10: Comparison with the progressive training of DCVC [R6] for TCM-small. All the models are trained for total 600k iterations with $\lambda=0.0483$.*
> | **Model (at 600k iterations)**    | **Distortion(PSNR)** | **Rate(bpp)** | **Test R-D Loss** |
> |------------------------------|----------------------|---------------|-------------------|
> | TCM-small (anchor model)     | 37.585               | 0.9006        | 1.4741            |
> | TCM-small + progressive training| 37.656            | 0.9412        | 1.4997            |
> | **TCM-small + AuxT**              | **37.834**               | **0.8865**        | **1.4243**           |
>
>
>
> We found that the progressive training approach results in a significant loss in rate-distortion performance and convergence speed. We speculate that this training strategy may be more suitable for inter-frame (P-frame) compression, which involves more complex modules such as motion compensation and motion vector compression. However, for intra-frame compression, joint R-D optimization performs better. In fact, DCVC also uses this R-D optimization for training intra-frame compression (see "Appendix C: Intra frame coding" of DCVC [R6]).
>
> **W3: Training time comparison with two-stage training strategy of the paper [R7]**
>
> Thank you for your suggestion. We notice in the paper [R7] that a total of 3.1M iterations were trained, where the first 2M iterations (about 65%) were used to train a small model with only the hyperprior and the remaining 1.1M iterations (about 35%) were used to train the model with a complex context model.
>
> For comparison, we allocated 1.3M iterations for training only the hyperprior in the case of ELIC and TCM-small, with the remaining 0.7M iterations dedicated to adding the complex context model in our 2M training iterations setting. The calculated training time comparison is presented in the table below. Our method demonstrates a significant reduction in training time, achieving decreases of 47%, 48%, and 47% compared to the anchor model for ELIC, TCM-small, and TCM-large, respectively. In contrast, the two-stage training strategy [R7] achieves only 11%, 30%, and 21% training time reductions for ELIC, TCM-small, and TCM-large, respectively.
>
>
> *Table R11: Training time comparison with two-stage training strategy of [R7].*
> | **Model**      | **#of Iterations** | **Training Time (GPU hours)** | **Training Time (GPU hours)** | **Training Time (GPU hours)** |
> |----------------|--------------------|-------------------------------|----|----|
> |                |                    | **ELIC** | **TCM-small** | **TCM-large** |
> | Anchor         | 2M                 | 143     | 240           | 330           |
> | Two-stage training [R7]| 1.3M+0.7M        | 127 (77+50)  (11\%$\downarrow$)| 167 (82+85)  (30\%$\downarrow$) | 261 (144+117)  (21\%$\downarrow)$|
> | **Ours**           | **1M**                 | **76    (47\%$\downarrow$ )** | **125        (48\%$\downarrow$ )**   | **175       ( 47\%$\downarrow$  )**  |
>
>
> [R6] Li, Jiahao, Bin Li, and Yan Lu. "Deep contextual video compression." Advances in Neural Information Processing Systems 34 (2021): 18114-18125.
>
> [R7] Zhu, Yinhao, Yang Yang, and Taco Cohen. "Transformer-based transform coding." International Conference on Learning Representations. 2022.

---

> > ### Comment · Reviewer_5Rti · 2024-11-23
> > **Feedback to authors**
> >
> > Thank you for your rebuttal. I will increase my rating accordingly.

---

> > > ### Author Response · Authors · 2024-11-23
> > >
> > > Thank you for your active feedback and engagement during the rebuttal process. We appreciate your endorsement!

---

### Official Review · Reviewer_xyeS · 2024-11-03

**Soundness:** 3
**Presentation:** 3
**Contribution:** 3
**Rating:** 8
**Confidence:** 4

**Summary:**

This paper identifies the slow convergence in training Learned Image Compression (LIC) models due to energy compaction issues and proposes a Linear Auxiliary Transform (AuxT) to address this. AuxT uses Wavelet-based Linear Shortcuts (WLSs) to improve feature decorrelation and energy modulation. Integrating AuxT into LIC models can accelerate training by 2-3 times and achieve a 1% reduction in BD-rate, improving both efficiency and performance.

**Strengths:**

(1) This paper explores the problem of slow convergence in LIC. Through a large number of experiments, the author designed a universal plug-in AuxT to solve this problem to a certain extent. I think the paper is interesting and novel.

(2) AuxT shows very strong generalization, achieving a significant reduction in training time and a small increase in performance in several classic LIC architectures (mbt2018,ELIC,TCM, etc.).

**Weaknesses:**

(1) In Subband-aware scaling of wavelet-based linear shortrcut (WLS), the authors appear to adopt simple linear scaling (reduction coefficients 1,0.5,0.5,0) for the four subbands LL,LH,HL,HH to modulate. Can you give further reasons for the use of these reduction coefficients, that is, it is proved through experiments that the reduction coefficients set in this way are optimal?

(2) The derivation of Loss Function seems to be a little short, I am not sure what $L_{orth}$ includes. Can you give the formula? Are they added on top of the original Rate-Distortion loss?

(3) In Table 1, what does Trainning time (-64%), (-40%) mean and to whom is it compared?

(4) TCM is a relatively new model in the field, and it seems that AuxT has relatively little performance improvement when applied on TCM. Can you further analyze the reason?

**Questions:**

Please see Weaknesses.

---

> ### Author Response · Authors · 2024-11-22
> **Response to Reviewer xyeS ( Part 1)**
>
> **W1: Scaling factor**
>
> In fact, \( 1, 0.5, 0.5, 0 \) represent the **initial values** of the scaling factors $ s_{LL} $, $ s_{LH} $, $ s_{HL} $, and $ s_{HH} $, respectively. These scaling factors are indeed **learnable parameters**, and are optimized with the entire network. The choice of initial values is motivated by the observed phenomenon of uneven energy modulation, where low-frequency components require more significant energy amplification than high-frequency components. Indeed, the initial value does not have significant impact on the overall performance. We have performed ablation studies with initial values set uniformly to 1 or randomly initialized. As shown in the table below, the overall performance is not sensitive to the initial value, and our uneven initialization is able to obtain a lowest test R-D loss.
>
> *Table 5: Effect of the initialized value of scaling factors. All models are trained for 600k iterations with $\lambda=0.0483$.*
> | **Models (at 600k iterations)** | **Distortion (PSNR)** | **Rate (bpp)** | **Test R-D Loss** |
> |----------------------------|-----------------------|----------------|-------------------|
> | Random                     | 37.817               | 0.8898         | 1.4288            |
> | 1, 1, 1, 1  (Uniform)  | 37.867               | 0.8944         | 1.4294            |
> | **1, 0.5, 0.5, 0 (Ours)** | **37.834**               | **0.8865**         | **1.4243**            |
>
> ---
>
> **W2: Loss Function**
>
> $L_{orth}$ represents the sum of orthogonality constraint terms across all OLPs, formulated as:
>
> $$L_{orth}=\sum_{\boldsymbol{W} \in \mathcal{W}} \left\|| \boldsymbol{W}^\top \boldsymbol{W} - \boldsymbol{I} \right\||_F^2,$$
>
> where $\boldsymbol{W}$ is the weight matrix of an OLP layer and $\mathcal{W}$ is the set of weight matrices for all OLPs.  The orthogonality constraint $\left\||\boldsymbol{W}^\top\boldsymbol{W}-\boldsymbol{I}\right\||_F^2$ was defined in Section 4.1 in the previously submitted manuscript to enforce the weight $\boldsymbol{W}$ to be an orthogonal matrix [R5].
>
>  This term $L_{orth}$ is added to the original R-D loss function defined in Equation (2) of the manuscript and the overall loss function for our method is :
>
> $$L_{overall} = L_{RD}+\lambda_{orth}L_{orth} = \mathcal{R} +\lambda \mathcal{D}+\lambda_{orth}L_{orth}.$$
>
> We have updated the our manuscript to clarify it.
>
> ---
>
> **W3: Unclear illustration of Table 1**
>
> We apologize for the unclear explanation. The relative values in Table 1 of the previously submitted manuscript compare the results with the anchor model trained for 2M iterations (the second row of each section). We have updated the revision to improve clarity.
>
> ---
>
> **W4: Performance gain for TCM**
>
> This is because TCM, as a relatively new state-of-the-art model, has already achieved a superior performance. As a result, further improving performance on this anchor model becomes increasingly challenging, leading to relatively limited gains. Additionally, it is important to note that our method was trained for only 1 million iterations, while the anchor model was trained for 2 million iterations. If training of our method were to continue, the gains could  increase further.
>
>
> [R5] Jiayun Wang, Yubei Chen, Rudrasis Chakraborty, and Stella X Yu. Orthogonal convolutional neural networks. In Proceedings of the IEEE/CVF conference on computer vision and pattern recognition, pp. 11505–11515, 2020.

---

> > ### Comment · Reviewer_xyeS · 2024-11-23
> >
> > Thanks to the authors for the response to the questions I raised in the review. I think the authors have addressed my concerns. I'd like to raise my score to accept.

---

> > > ### Author Response · Authors · 2024-11-23
> > >
> > > Thank you for your active feedback and engagement during the rebuttal process. We appreciate your endorsement!

---

### Official Review · Reviewer_DW5J · 2024-11-08

**Soundness:** 4
**Presentation:** 4
**Contribution:** 4
**Rating:** 8
**Confidence:** 5

**Summary:**

This paper introduces a linear auxiliary transform (AuxT) to address the slow convergence issue in learned image compression (LIC). Specifically, this paper analyzes and designs the method based on feature decorrelation and uneven energy modulation. Experimental results demonstrate that, compared to traditional LIC methods, AuxT accelerates model training by 2 to 3 times and achieves an average 1% BD-rate reduction, improving convergence and rate-distortion performance.

**Strengths:**

1. The paper is well-written.
2. This paper introduces a valuable new research direction. Accelerating convergence in training learned image codecs is indeed an underexplored area and is both novel and important for this field.
3. The idea of analyzing the training process from the perspective of energy compaction is interesting.
4. The experiment results are extensive and complete, showing the effectiveness of the proposed method.

**Weaknesses:**

1. This paper appears to focus solely on cases where learned image codecs do not incorporate a context model. However, including a context model could potentially enhance coding performance by leveraging correlations between latents. However, this paper primarily aims to decorrelate. It remains unclear whether the proposed method is applicable in scenarios with a context model.
2. The motivation of adding additional branch is unclear. Is it possible to merge the operation of the proposed method (DWT / IDWT + Sub-band-aware scaling + OLP + orthogonality constraint) to $g_a$ and $g_s$?
3. In Section A.2, this paper compares the proposed method with two alternatives that substitutes the sub-sampling and up-sampling layers with DWT and IDWT to highlight the effectiveness of introducing a bypass auxiliary transform. However, the comparison is not entirely fair, as the proposed method also includes OLP layers and orthogonality constraints.
4. The appendix of this paper compares the proposed method with an energy compaction penalty. However, the paper “Towards Efficient Image Compression Without Autoregressive Models (NeurIPS'23)” also addresses latent decorrelation for learned image compression, but this paper does not discuss or compare it.

**Questions:**

1. The input to the WLAS in the proposed method is either the coded image x or the output from the previous layer of WLAS. I t would be interesting to compare this with using the intermediate features from the analysis transform at the corresponding resolution as input for the 2nd, 3rd, and 4th layers of WLAS. Similarly, for iWLAS, the input could be the intermediate features from the synthesis transform at the corresponding resolution.
2. Are $s_{ll}, s_{lh}, s_{hl}, s_{hh}$ trainable? Are they shared across all WLAS and iWLAS?
3. Is there an explanation for why employing more complex wavelets, such as the Daubechies wavelet db4 and biorthogonal wavelet bior2.2, performs worse than the simpler Haar wavelet?
4. In the ablation study “Effect of the linearity of AuxT,” it is mentioned that the 1x1 convolution layer W in OLP is placed before the added non-linear (ReLU or GDN) layer in comparison. However, it is unclear whether the orthogonality constraint is applied before or after the non-linear layer.
5. In Fig.1, it would be better to explain how energy is measured.
6. Cross-ref of figures mentioned in Section 5.4 should be wrong.
7. In Fig. 8, [A4] DWT -> Conv should be [A3] DWT -> Conv.

---

> ### Author Response · Authors · 2024-11-22
> **Response to Reviewer DW5J ( Part 1)**
>
> ---
>
> **W1: On the context model**
>
> Thank you for your advice. It is noted that we did evaluate the proposed method on LIC models with diverse context models, as shown in Table 1 of the previously submitted manuscript. The context models include the space-channel context model in ELIC, the channel-wise context model in STF, and the transformer-based channel-wise context model in TCM.  We found that the proposed method benefits all these methods and achieves BD-rate reduction from 0.7\% to 2.5\% when incorporated into these models.  We have revised the manuscript to clarify the compatibility of the proposed method with diverse context models.
>
> ---
>
> **W2: Reason for introducing an additional branch**
>
> The additional branch helps accelerate convergence  in training LIC models with nonlinear transforms. The main branch (*i.e.*, nonlinear transform) and the additional branch (*i.e.*, the proposed AuxT) can work in parallel to extract fine-grained and coarse information from the images. However, this cannot be achieved when AuxT is employed as part of a sequential model.   Additionally, we have conducted a study on merging the proposed WLS/iWLS into the nonlinear transform by substituting the sub-sampling and up-sampling layers of TCM with the proposed WLS/iWLS (*i.e.*, DWT/IDWT + Sub-band-aware scaling + OLP + orthogonality constraint).  The model variants used in the experiments are as follows:
>
> - (a) TCM-small (anchor model)
> - (b) Substituting all the sub-sampling and up-sampling layers of TCM-small with DWT/IDWT (single branch)
> - (c) Substituting all the sub-sampling and up-sampling layers of TCM-small with the proposed WLS/iWLS (single branch)
> - (d) TCM-small + AuxT (two parallel branches of our method)
>
>
> The R-D results are shown in the table below and the convergence curves are provided in Fig.9 of the revised manuscript.
>
>
> *Table R1: Effect on merging WLS/iWLS to $g_a$ and $g_s$. All the models are trained for 600k iterations with $\lambda=0.0483$.*
> | **Models** | **#Params (M)** | **Distortion (PSNR)** | **Rate (bpp)** | **Test R-D Loss** |
> |------------|----------------|----------------------|----------------|------------------|
> | (a)        | 45.2          | 37.585              | 0.9006         | 1.4741           |
> | (b)        | 51.5          | 37.675              | 0.9031         | 1.4613           |
> | (c)        | 52.9          | 37.520              | 0.9074         | 1.4886           |
> | **(d)**        | **45.8**          | **37.834**               | **0.8865**          | **1.4243**            |
>
> Compared to introducing additional branches, merging WLS/iWLS into $g_a$ and $g_s$ causes degraded  R-D performance.  This may occur because these additional modules and constraints hinder the nonlinear transforms from effectively learning latent representations for fine details. This further highlights the importance of the additional parallel branch design in accelerating convergence.
>
> ---
>
> **W3: Substituting the sub-sampling and up-sampling layers with DWT and IDWT**
>
> Thank you for your valuable suggestion. As introduced in the response to W2, we have conducted an additional study on substituting the sub-sampling and up-sampling layers of TCM with the full wavelet-based shortcuts (*i.e.*, DWT/IDWT + Sub-band-aware scaling + OLP + orthogonality constraint).   The results indicate that incorporating sub-band-aware scaling, OLP, and orthogonality constraints leads to a performance drop compared to using only DWT/IDWT.

---

> ### Author Response · Authors · 2024-11-22
> **Response to Reviewer DW5J ( Part 2)**
>
> ---
>
> **W4: Comparison with the correlation loss of NIPS2023**
>
> The proposed method differs from the correlation loss [R1] in both objectives and methodology. [R1]  aims to enhance the decorrelating ability of nonlinear transforms without relying on a context model. It designs a metric for evaluating the neighborhood correlations of latent representations and uses it as a penalty in the loss function. Different from [R1], we focus on accelerating the convergence of training nonlinear transforms by learning coarse decorrelated representations with an easy-to-learn auxiliary transform.
> It is worth mentioning that the correlation loss and our method are orthogonal and can be combined to complementary each other to further enhance the network's decorrelating ability and accelerate convergence. We have discussed this paper in the revised appendices.
>
> We have attempted to compare the proposed method with [R1]. However, the codes of the method developed in [R1] are not yet publicly available and we suffer from unstable convergence (*i.e.*, NaN loss in training) when reproducing it. We strictly followed the methodology and settings described in the original paper, as well as the hyperparameters provided in the appendix. In detail, we set the training iterations to 600K for comparison and applied the correlation loss [R1] to diverse models (*e.g.*, TCM-small and mbt2018-mean) and conducted experiments on different bitrate points (*i.e.*, $\lambda= 0.0018$ for low bitrate and  $\lambda = 0.0483$ for high bitrate). We consistently observed a training collapse with the loss becoming NaN.  Moreover, prior to the collapse, we did not observe any significant performance improvements compared to the anchor model. The comparison between the proposed method (before collapse) and the anchor are depicted in the following tables.
>
> *Table R2: Comparison with correlation loss [R1] using $\lambda=0.0483$. Training with the correlation loss collapses at 410k iterations.*
> | **Models (at 410k iterations)**      | **Distortion (PSNR)** | **Rate (bpp)** | **Test R-D Loss** |
> |----------------------------------|-----------------------|----------------|-------------------|
> | TCM-small (anchor model)         | 37.051               | 0.9045         | 1.5536            |
> | TCM-small + correlation loss     | 37.134               | 0.9119         | 1.5527            |
> | **TCM-small + AuxT**                 | **37.696**               | **0.9123**         | **1.4661**            |
>
> *Table R3: Comparison with correlation loss [R1] using $\lambda=0.0018$. Training with the correlation loss collapses at 100k iterations.*
> | **Models (at 100k iterations)**       | **Distortion (PSNR)** | **Rate (bpp)** | **Test R-D Loss** |
> |----------------------------------|-----------------------|----------------|-------------------|
> | mbt2018mean (anchor model)       | 27.554               | 0.1327         | 0.3638            |
> | mbt2018mean + correlation loss   | 27.522               | 0.1308         | 0.3640            |
> | **mbt2018mean + AuxT**               | **27.749**               | **0.1329**         | **0.3542**            |
>
>
>
>
> **[R1]** Ali, Muhammad Salman, et al. "Towards efficient image compression without autoregressive models." *Advances in Neural Information Processing Systems*, 36 (2024).
>
> ---
>
> **Q1: Input of WLS**
>
> Thank you for your valuable advice. We have performed experiments where the inputs of WLS/i-WLS were replaced with the intermediate features of $g_a$ or $g_s$, denoted as  AuxT*. This modification means that WLSs no longer form a standalone bypass transform by sequentially stacking but instead act as shortcuts at each intermediate stage of the nonlinear transform separately. The results are shown in the table below. We find that this modification causes degraded rate-distortion performance. The results further demonstrate the necessity of introducing an additional branch for accelerated convergence of LIC.
>
> *Table R4: Effect of the input of WLS. All the models are trained for 600k iterations with $\lambda=0.0483$.*
> | **Models (at 600k iterations)**        | **Distortion (PSNR)** | **Rate (bpp)** | **Test R-D Loss** |
> |-----------------------------------|-----------------------|----------------|-------------------|
> | TCM-small (anchor model)          | 37.585               | 0.9006         | 1.4741            |
> | **TCM-small + AuxT**                  | **37.834**               | **0.8865**         | **1.4243**            |
> | TCM-small + AuxT*                 | 37.774               | 0.8957         | 1.4404            |
>
>
>
> ---
>
> **Q2: Are scaling factors trainable?**
>
> Yes, the scaling factors are learnable. Each WLS/iWLS employs a set of scaling factors and they are jointly optimized with other parts of the model. We have updated Section 4.1 in the revised manuscript to clarify it.
>
> ---

---

> ### Author Response · Authors · 2024-11-22
> **Response to Reviewer DW5J ( Part 3)**
>
> ---
>
> **Q3: Why is a simple wavelet more effective?**
>
>
> The Haar wavelet is the simplest wavelet that is both orthogonal and symmetric, ensuring a linear phase property, which minimizes phase distortion and improves compression performance [R3, R4]. In contrast, the db4 wavelet lacks the symmetry of the Haar wavelet, potentially introducing phase distortion and necessitating the use of discontinuous periodic boundary extensions, which can introduce high-frequency artifacts in the reconstructed image [R5]. Although the bior2.2 wavelet is symmetric, it is not orthogonal and results in a loss of overall orthogonality when applied in our AuxT. This causes degraded performance. Moreover, the Haar wavelet is more computation-efficient, since it has a short support and requires less processing power and time compared to wavelets with longer supports, such as db4 and bior2.2.
>
> ---
>
> **Q4: Effect of the linearity of AuxT**
>
> The used orthogonality constraint cannot regularize the nonlinear activation layer, as it is designed to constrain the weights $\boldsymbol{W}$ of the OLP, **meaning it is implemented before the non-linear layer**. However, the non-linear layer disrupts the overall orthogonality and linearity, which may cause degraded performance and slower convergence speed.
>
> ---
>
> **Q5: Figure 1 and other typos**
>
> Thank you for your kind reminder. The energy of each channel is computed using the $L_2$ norm in Fig.1. We have updated the caption of Fig.1 and corrected the typos you mentioned.
>
> **[R2]** Soman, Anand K., P. P. Vaidyanathan, and Truong Q. Nguyen. "Linear phase paraunitary filter banks: Theory, factorizations and designs." IEEE transactions on signal processing 41.12 (1993): 3480-3496.
>
> **[R3]** Stanhill, David, and Yehoshua Y. Zeevi. "Two-dimensional orthogonal filter banks and wavelets with linear phase." IEEE transactions on signal processing 46.1 (1998): 183-190.
>
> **[R4]** Belzer, Benjamin, J-M. Lina, and John Villasenor. "Complex, linear-phase filters for efficient image coding." IEEE Transactions on Signal Processing 43.10 (1995): 2425-2427.

---

> ### Comment · Reviewer_DW5J · 2024-11-25
>
> Thank you for your detailed response addressing my questions. I will hold the same score.

---

> > ### Author Response · Authors · 2024-11-26
> >
> > Thank you for your engagement during the rebuttal process. We appreciate your effort and feedback.

---

### Author Response · Authors · 2024-11-22
**General Response**

We sincerely thank  all the reviewers for their valuable feedback to improve the quality of our manuscript. We have updated the manuscript with the revised contents highlighted in blue. The main changes are summarized as below.

 **Abstract**

- We have refined the expressions to more accurately describe our contributions.

 **Captions of Table and Figure**

- We have updated the captions of Figures 1 and 5 for clearer illustration.
- We have included descriptions for the relative values in the caption of Table 1 to improve readability.

 **Method**

- We have clarified the demonstration of subband-aware scaling in Section 4.
- We have clarified the loss function in Section 4 with detailed formulation.

 **Appendix**

- We have included the related works on *Decorrelation for LIC* and *Training Acceleration for Neural Networks* to provide a broader context.
- We have included new experimental results in Figure 9 for a more appropriate ablation study.

---

### Meta-Review · Area_Chair_wsc5 · 2024-12-16

**Metareview:**

The paper addresses slow training in learned image compression methods. Their approach is to augment the analysis and synthesis transforms with parallel branches based on wavelets, resulting in a 50% decrease in training time. It is demonstrated that this method generalizes to several known architectures. The authors also provide an interesting analysis of their method in terms of energy compaction.

**Additional Comments On Reviewer Discussion:**

The review of this paper was straight forward – the reviewers mainly had technical questions, which were clarified to their satisfaction. There were no major weaknesses noted.

---

### Decision · Program_Chairs · 2025-01-22

Accept (Spotlight)